# Senescent cells inhibit mouse myoblast differentiation via the SASP-lipid 15d-PGJ₂ mediated modification and control of HRas

Swarang Sachin Pundlik[1,2], Alok Barik[1], Ashwin Venkateshvaran[1], Snehasudha Subhadarshini Sahoo[1,3], Mahapatra Anshuman Jaysingh[4,5], Raviswamy GH Math[6], Heera Lal[1,2], Maroof Athar Hashmi[1,2], Arvind Ramanathan[1]*

[1]Metabolic Regulation of Cell Fate (RCF), Institute for Stem Cell Science and Regenerative Medicine (InStem), Bangalore Life Science Cluster, Bengaluru, India; [2]Manipal Academy of Higher Education (MAHE), Manipal, India; [3]University of North Carolina at Chapel Hill, Chapel Hill, United States; [4]Department of Biological Sciences, Indian Institute of Science Education and Research Kolkata (IISER-K), Mohanpur, India; [5]Division of Biology and Biomedical Sciences, Washington University in St Louis, St Louis, United States; [6]National Centre for Biological Sciences (NCBS), Bengaluru, India

*For correspondence:
arvind@instem.res.in

Competing interest: The authors declare that no competing interests exist.

**Abstract** Senescent cells are characterized by multiple features such as increased expression of senescence-associated β-galactosidase activity (SA β-gal) and cell cycle inhibitors such as p21 or p16. They accumulate with tissue damage and dysregulate tissue homeostasis. In the context of skeletal muscle, it is known that agents used for chemotherapy such as Doxorubicin (Doxo) cause buildup of senescent cells, leading to the inhibition of tissue regeneration. Senescent cells influence the neighboring cells via numerous secreted factors which form the senescence-associated secreted phenotype (SASP). Lipids are emerging as a key component of SASP that can control tissue homeostasis. Arachidonic acid-derived lipids have been shown to accumulate within senescent cells, specifically 15d-PGJ₂, which is an electrophilic lipid produced by the non-enzymatic dehydration of the prostaglandin PGD₂. This study shows that 15d-PGJ₂ is also released by Doxo-induced senescent cells as an SASP factor. Treatment of skeletal muscle myoblasts with the conditioned medium from these senescent cells inhibits myoblast fusion during differentiation. Inhibition of L-PTGDS, the enzyme that synthesizes PGD₂, diminishes the release of 15d-PGJ₂ by senescent cells and restores muscle differentiation. We further show that this lipid post-translationally modifies Cys184 of HRas in C2C12 mouse skeletal myoblasts, causing a reduction in the localization of HRas to the Golgi, increased HRas binding to Ras Binding Domain (RBD) of RAF Kinase (RAF-RBD), and activation of cellular Mitogen Activated Protein (MAP) kinase–Extracellular Signal Regulated Kinase (Erk) signaling (but not the Akt signaling). Mutating C184 of HRas prevents the ability of 15d-PGJ₂ to inhibit the differentiation of muscle cells and control the activity of HRas. This work shows that 15d-PGJ₂ released from senescent cells could be targeted to restore muscle homeostasis after chemotherapy.

## eLife assessment

This manuscript outlines an interaction between senescence-related 15d-PGJ2 and the proliferation and differentiation of myoblasts, with potential implications for muscle health. This manuscript is **useful** in understanding the role of lipid metabolite 15d-PGJ2 in myoblast proliferation

and differentiation. However, in its current form, the manuscript is **incomplete** as there are several concerns in the statistical analysis, lack of clarity on the mechanistic details, and concerns about the use of an immortalized C2C12 myoblasts cell line to draw major conclusions related to senescence-associated secreted phenotype.

## Introduction

Senescent cells are important drivers of aging and damage-associated loss of tissue homeostasis (*Childs et al., 2015*). Anti-cancer chemotherapy presents an important context where treatment with chemotherapeutics such as Doxorubicin (Doxo) causes widespread cellular senescence which inhibits tissue homeostasis and regeneration, including in skeletal muscles (*Francis et al., 2022*). It has been shown that Doxo causes systemic inflammation and leads to the emergence of senescent cells across tissues (*Di Leonardo et al., 1994*; *Hu and Zhang, 2019*; *Robles and Adami, 1998*). Senescent cells negatively affect tissue homeostasis and regeneration by releasing factors including proteins like growth factors, matrix metalloproteases, cytokines, and chemokines, and small molecules like fatty acid derivatives (*Campisi, 2005*; *Coppé et al., 2010*; *Dilley et al., 2003*; *Krtolica et al., 2001*; *Parrinello et al., 2005*; *Shelton et al., 1999*; *Yang et al., 2006*). The release of these factors from senescent cells is called the senescence-associated secretory phenotype (SASP). It is expected that these SASP factors and their mechanisms of action will vary depending on cellular and tissue contexts. Identifying SASP factors and their underlying mechanistic targets will be critical for building an understanding of how senescent cells control tissue homeostasis (*Coppé et al., 2010*; *Davalos et al., 2010*). Lipids are a less explored family of SASP factors, and it is important to understand how they affect tissue regeneration (*Hamsanathan and Gurkar, 2022*). We have previously shown that senescent cells have increased intracellular levels of prostaglandin 15d-PGJ$_2$ (*Wiley et al., 2021*), a non-enzymatic dehydration product of prostaglandin PGD$_2$ (*Shibata et al., 2002*). In the context of skeletal muscle, PGD$_2$ and 15d-PGJ$_2$ have been shown to negatively regulate muscle differentiation via mechanisms that do not depend on a cognate receptor (*Hunter et al., 2001*; *Veliça et al., 2010*). Here, we study the role of 15d-PGJ$_2$ as a member of the SASP and identify the mechanisms by which it might negatively affect muscle regeneration. 15d-PGJ$_2$ has been previously shown to covalently modify multiple proteins like MAPK1, MCM4, EIF4A-I, PKM1, GFAP, etc. in endothelial and neuronal cells (*Marcone and Fitzgerald, 2013*; *Yamamoto et al., 2011*). 15d-PGJ$_2$ was shown to be covalently modifying HRas in NIH3T3, Cos1, and IMR90 cell lines (*Oliva et al., 2003*; *Wiley et al., 2021*). We further studied HRas as an important target that might mediate the effects of 15d-PGJ$_2$ on muscle differentiation via covalent modification. We investigated HRas as a possible effector of 15d-PGJ$_2$ because (1) HRas belongs to the Ras superfamily of small molecule GTPases and is a known regulator of key cellular processes (*Davis et al., 1983*; *Harvey, 1964*; *Kirsten and Mayer, 1967*; *Vetter and Wittinghofer, 2001*), (2) constitutively active HRas mutant (HRas V12) has been shown to inhibit the differentiation of myoblasts by inhibiting MyoD and Myogenin expression (*Konieczny et al., 1989*; *Lassar et al., 1989*; *Olson et al., 1987*; *van der Burgt et al., 2007*), and (3) downstream signaling of HRas is important for muscle homeostasis as skeletal and cardiac myopathies are observed in individuals carrying constitutively active mutants of HRas (*Engler et al., 2021*; *Konieczny et al., 1989*; *Lee et al., 2010*; *Olson et al., 1987*; *Scholz et al., 2009*; *van der Burgt et al., 2007*). HRas is highly regulated by lipid modifications, it undergoes reversible palmitoylation and de-palmitoylation at C-terminal cysteines, which regulate the intracellular distribution and activity of HRas (*Gutierrez et al., 1989*; *Lu and Hofmann, 1995*; *Rocks et al., 2005*). In this study, we show that 15d-PGJ$_2$ is synthesized and released by senescent myoblasts upon treatment with Doxo. 15d-PGJ$_2$, taken up by the myoblasts, covalently modifies HRas at cysteine 184 and activates it. We also show that previously reported inhibition of differentiation of myoblasts by 15d-PGJ$_2$ depends on HRas C-terminal cysteines, notably cysteine 184. This study provides a mechanism by which prostaglandins secreted as SASP inhibit the differentiation of myoblasts, affecting muscle homeostasis in patients undergoing chemotherapy.

## Results

### Doxo treatment induces senescence in mouse skeletal muscles and C2C12 mouse myoblasts

Doxo-mediated DNA damage has been shown to induce senescence in cells (*Di Leonardo et al., 1994*; *Hu and Zhang, 2019*; *Robles and Adami, 1998*). Therefore, we injected B6J mice intraperitoneally with Doxo (5 mg/kg) every 3 days for 9 days and observed induction of DNA damage-mediated senescence in hindlimb skeletal muscles (*Figure 1—figure supplement 1A*). We observed an increase in the expression of p21 and increased nuclear levels of the DNA damage marker γH2A.X in mouse Gastrocnemius muscles (*Figure 1A, B*). We also observed a significant increase in the mRNA levels of known senescence markers (*Cdkn1a* and *Cdkn2a*), SASP factors (*Cxcl1*, *Cxcl2*, *Tnfa*, *Il6*, and *Tgfb1*) in skeletal muscles of mice treated with Doxo compared to that of mice treated with saline (*Figure 1C*). These observations suggest that there is induction of senescence in skeletal muscles of mice upon treatment with Doxo.

Treatment with Doxo induces senescence in cancer cells like HeLa cells (*Hu and Zhang, 2019*). We treated MCF7 human breast adenocarcinoma cells with Doxo (20 and 50 nM) and observed flattened cell morphology (*Figure 1—figure supplement 1C*), an increase in the nuclear accumulation of cell cycle inhibitor p21 (*Figure 1—figure supplement 1D*), an increase in the size of nuclei (*Figure 1—figure supplement 1E*), and an increase in the levels of senescence-associated β-galactosidase activity (SA β-gal) (*Figure 1—figure supplement 1F*). These observations suggest that MCF7 cells undergo senescence upon treatment with Doxo.

C2C12 cells have been shown to undergo senescence after DNA damage, as assessed by an increase in the levels of SA β-gal and known markers of SASP (IL1α, IL6, CCL2, CXCL2, and CXCL10) (*Moiseeva et al., 2023*). We treated C2C12 myoblasts with Doxo (150 nM) and observed a significant increase in the size of the nuclei (*Figure 1D*), flattened cell morphology with an increase in the cell size (*Figure 1E*), a significant increase in the mRNA levels of cell cycle inhibitor Cdkn1a and SASP factors Il6 and Tgfb (*Figure 1F*), a significant increase in the protein levels of p21 (*Figure 1G*), and an increase in the levels of SA β-gal (*Figure 1H*) in C2C12 cells treated with Doxo. These observations suggest that C2C12 cells undergo senescence upon treatment with Doxo.

### Doxo-mediated senescence induces synthesis and release of 15d-PGJ$_2$ in C2C12 myoblasts and mouse skeletal muscle

Synthesis of prostaglandins by senescent cells has previously been reported (*Wiley et al., 2021*; *Wiley and Campisi, 2021*). Specifically, levels of PGD$_2$ and its metabolite 15d-PGJ$_2$ have been shown to be significantly increased in senescent cells. Therefore, we measured the levels of mRNA of enzymes involved in the synthesis of PGD$_2$/15d-PGJ$_2$ (*Ptgs1*, *Ptgs2*, and *Ptgds*), in the gastrocnemius muscle of mice after treatment with Doxo. We observed a significant increase in the mRNA levels of *Ptgs1*, *Ptgs2*, and *Ptgds* enzymes in the skeletal muscle of mice treated with Doxo (*Figure 1C*). We also observed a time-dependent increase in the mRNA levels of *Ptgs1*, *Ptgs2*, *Ptgds*, and *Ptges* enzymes in C2C12 cells treated with Doxo compared to Day 0 (*Figure 1I*). Expression of enzyme *Ptges* was elevated on Day 4, whereas the expression of prostaglandin D synthase (*Ptgds*) increased only after Day 8, reaching maximum expression on Day 12. These observations suggest an increase in the synthesis of prostaglandins in senescent cells.

15d-PGJ$_2$ is a non-enzymatic dehydration product of PGD$_2$ (*Shibata et al., 2002*). We observed an increase in the mRNA levels of synthetic enzymes of 15d-PGJ$_2$ in senescent C2C12 cells. Therefore, we measured the levels of 15d-PGJ$_2$ released by senescent C2C12 cells using targeted mass spectrometry (*Figure 1—figure supplement 1G–J*). The concentration of 15d-PGJ$_2$ was quantified by monitoring the transition of the *m/z* of ions from 315.100 ➜ 271.100 using a SCIEX 5500 mass spectrometer. We plotted a standard curve using purified 15d-PGJ$_2$ (*Figure 1—figure supplement 1J*) to quantify the concentration of 15d-PGJ$_2$. We used the representative peaks from the conditioned medium collected from C2C12 cells incubated in 0.2% serum medium for 3 days (quiescent cells) and C2C12 cells treated with Doxo (150 nM) (senescent cells) to measure the concentration of 15d-PGJ$_2$ released by quiescent cells or senescent C2C12 cells. We observed a significant increase (~40-fold) in the concentration of 15d-PGJ$_2$ in the conditioned medium from senescent cells as compared to that in quiescent cells (*Figure 1J*, *Figure 1—source data 4*). This suggests that senescent C2C12 cells release 15d-PGJ$_2$ in the medium.

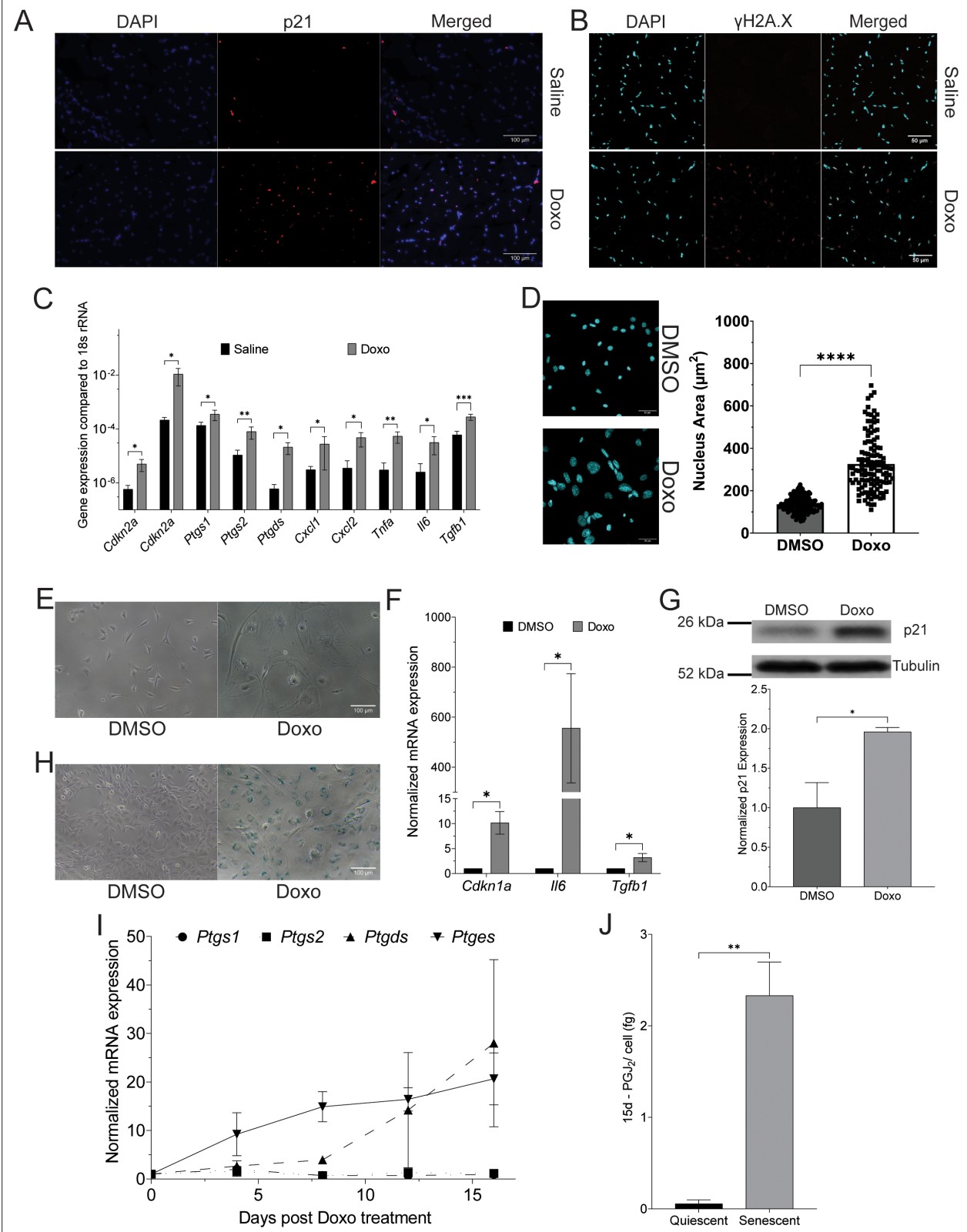

**Figure 1.** Treatment with Doxorubicin (Doxo) induces senescence in vitro and in vivo and leads to the release of prostaglandin 15d-PGJ$_2$ by senescent cells. (**A**) Expression and localization of tumor suppressor protein p21, measured by immunofluorescence, in hindlimb skeletal muscles of mice after 11 days of treatment with Doxo (5 mg/kg) or Saline (N = 3). (**B**) Representative confocal micrograph of expression of γH2A.X in the gastrocnemius muscle of mice treated with Doxo (5 mg/kg) or Saline (N = 3). (**C**) Expression of mRNAs of senescence markers (*Cdkn2a* and *Cdkn1a*), senescence-

*Figure 1 continued on next page*

*Figure 1 continued*

associated secreted phenotype (SASP) factors (*Cxcl1*, *Cxcl2*, *Tnfa*, *Il6*, and *Tgfb1*), and enzymes involved in the biosynthesis of prostaglandin $PGD_2$/15d-$PGJ_2$ (*Ptgs1*, *Ptgs2*, and *Ptgds*), measured by quantitative polymerase chain reaction (qPCR), in hindlimb skeletal muscles of mice after 11 days of treatment with Doxo (5 mg/kg) or Saline ($N = 4$). (**D**) A representative confocal micrograph and a scatter plot of the nuclear area of C2C12 myoblasts, measured by immunofluorescence, after 16 days of treatment with Doxo (150 nM) or Dimethyl Sulphoxide (DMSO) ($N = 3$). (**E**) A representative widefield micrograph of cell morphology in C2C12 myoblasts after 16 days of treatment with Doxo (150 nM) or DMSO ($N = 3$). (**F**) Expression of mRNA of cell cycle inhibitor *Cdkn1a* and SASP factors (*Il6* and *Tgfb1*), measured by qPCR, in C2C12 myoblasts after 16 days of treatment with Doxo (150 nM) or DMSO ($N = 3$). (**G**) Expression of cell cycle inhibitor p21, measured by immunoblot, in C2C12 myoblasts after 16 days of treatment with Doxo (150 nM) or DMSO ($N = 3$). (**H**) Activity of senescence-associated β-galactosidase (SA β-gal), measured by X-gal staining at pH ~6, in C2C12 myoblasts after 16 days of treatment with Doxo (150 nM) or DMSO ($N = 3$). (**I**) Expression of mRNAs of prostaglandin biosynthetic enzymes, measured by qPCR, in C2C12 myoblasts after treatment with Doxo (150 nM) or DMSO ($N = 4$). (Statistical significance was tested using Dunnett's multiple comparisons test (*Figure 1—source data 3*).) (**J**) Concentration of 15d-$PGJ_2$ released from quiescent or senescent C2C12 cells ($N = 3$). (The Standard Deviation between replicates was plotted as error bars. Statistical significance was tested by the two-tailed Student's *t*-test ns = $p > 0.05$, *$p < 0.05$, **$p < 0.01$, ***$p < 0.001$, ****$p < 0.0001$.)

The online version of this article includes the following source data and figure supplement(s) for figure 1:

**Source data 1.** Uncropped and labelled gels for *Figure 1*.

**Source data 2.** Raw unedited gels for *Figure 1*.

**Source data 3.** Dunnett's multiple comparison test for the time-dependent expression of prostaglandin biosynthesis enzymes.

**Source data 4.** 15d-$PGJ_2$ concentration in the conditioned medium using mass spectrometry.

**Figure supplement 1.** Treatment with Doxorubicin (Doxo) induces senescence in vivo and in vitro and induces release of eicosanoid prostaglandin 15d-$PGJ_2$.

## Prostaglandin $PGD_2$ and its metabolites in the conditioned medium of senescent cells inhibit the differentiation of C2C12 myoblasts

15d-$PGJ_2$ (the final non-enzymatic dehydration product of $PGD_2$) has been shown to inhibit the differentiation of myoblasts (*Hunter et al., 2001*). We observed the release of 15d-$PGJ_2$ by senescent cells, showing that 15d-$PGJ_2$ is an SASP factor (*Figure 1J*). Conditioned medium of senescent cells inhibits the differentiation of myoblasts in myotonic dystrophy type 1 (*Conte et al., 2023*). Therefore, we tested whether 15d-$PGJ_2$, the terminal dehydration product of $PGD_2$, is required for the inhibitory effect of SASP on the differentiation of myoblasts. We treated C2C12 myoblasts with the conditioned medium of senescent cells or senescent cells treated with 30 µM of AT-56 (a well-characterized inhibitor of prostaglandin D synthase (PTGDS)) (*Hu et al., 2022*; *Hu et al., 2023a*; *Hu et al., 2023b*; *Irikura et al., 2009*) and measured the differentiation of myoblasts by calculating the fusion index. We observed a significant decrease (~20%) in the fusion index of the C2C12 myoblasts treated with the conditioned medium of senescent cells (*Figure 2A*), suggesting that SASP factors decrease the differentiation of myoblasts. This decrease in the inhibition was rescued in myoblasts treated with the conditioned medium of senescent cells treated with AT-56 (*Figure 2A*). This suggests that prostaglandins $PGD_2$/15d-$PGJ_2$ released by senescent cells as SASP factors can inhibit the differentiation of myoblasts.

## 15d-$PGJ_2$ inhibits the proliferation and differentiation of mouse and human myoblasts

15d-$PGJ_2$ has been shown to affect the proliferation of cancer cell lines, both positively and negatively (*Chen et al., 2003*; *Choi et al., 2020*; *Slanovc et al., 2024*; *Yen et al., 2014*). We measured the effect of 15d-$PGJ_2$ on the proliferation of C2C12 myoblasts. We treated C2C12 myoblasts with 15d-$PGJ_2$ (10 µM) or DMSO in DMEM (Dulbecco's Modified Eagle Medium)10% serum medium for 72 hr and observed a significant decrease in the proliferation of C2C12 cells after treatment with 15d-$PGJ_2$ (*Figure 2B*). The doubling time of C2C12 cells was also increased upon treatment with 15d-$PGJ_2$ (57.24 hr) compared to DMSO (13.76 hr). This suggests that 15d-$PGJ_2$ decreases the proliferation of C2C12 myoblasts.

We measured the differentiation of C2C12 mouse and primary human myoblasts after treatment with 15d-$PGJ_2$. To rule out the toxic effects of 15d-$PGJ_2$ on cell physiology, we treated C2C12 cells with 15d-$PGJ_2$ (1, 2, 4, 5, and 10 µM) in the C2C12 differentiation medium and measured the viability of cells after 24 hr of treatment, using an MTT viability assay. We judged that 15d-$PGJ_2$ was not cytotoxic up to 5 µM in the C2C12 differentiation medium (*Figure 2—figure supplement 1A*). Based

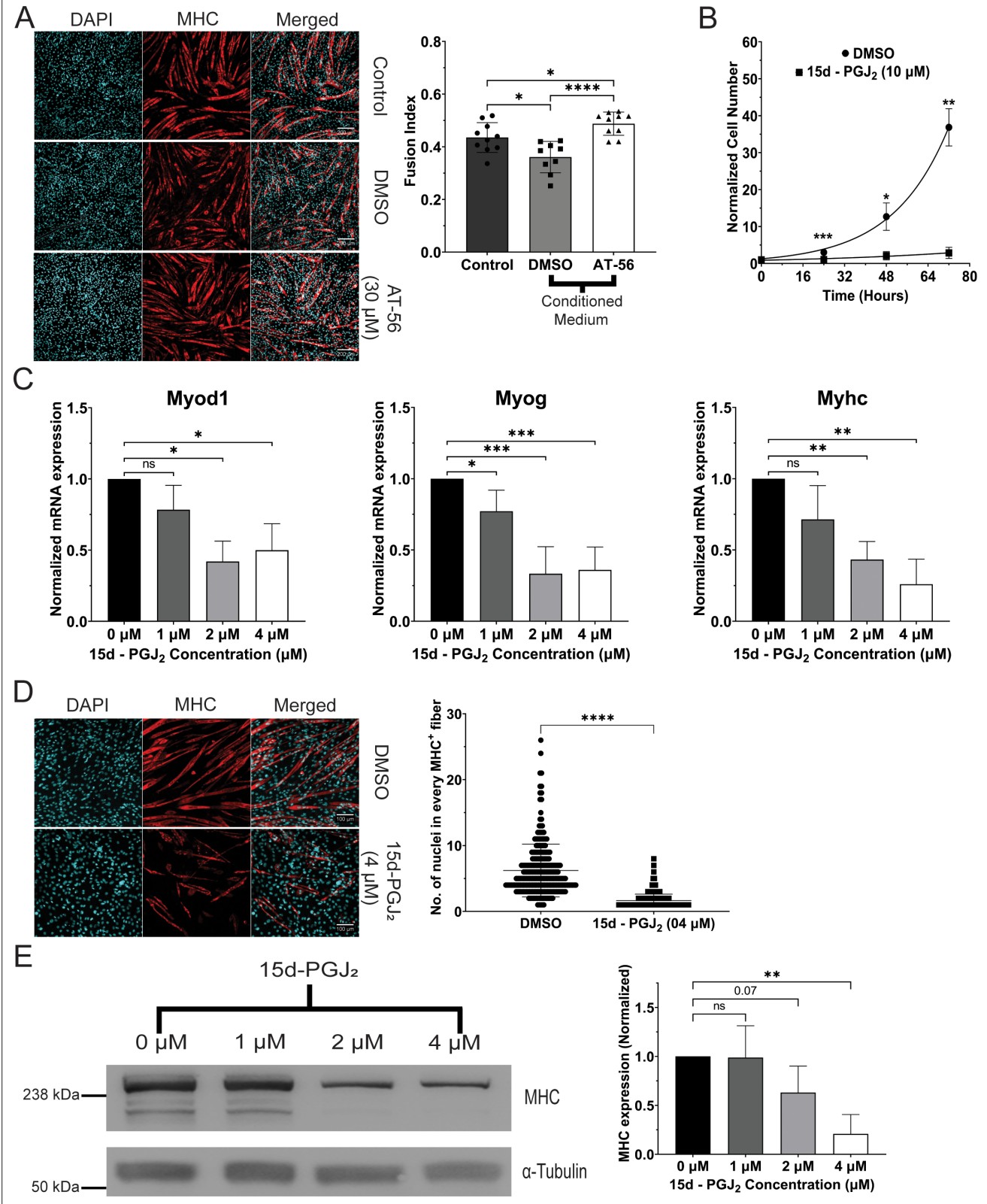

**Figure 2.** 15d-PGJ$_2$ inhibits differentiation of myoblasts. (**A**) Expression of Myosin Heavy Chain (MHC) protein and the fusion of myoblasts in myotubes, measured by immunofluorescence, after treatment with conditioned medium of senescent cells treated with prostaglandin D synthase (PTGDS) inhibitor AT-56 (30 µM) or DMSO. (**B**) Normalized number of C2C12 myoblasts treated with 15d-PGJ$_2$ (10 µM) or DMSO (*N* = 3). (**C**) Expression of mRNAs of markers of differentiation (*Myod1*, *Myog*, and *Myhc*), measured by qPCR, in C2C12 myoblasts treated with 15d-PGJ$_2$ (1, 2, or 4 µM) or DMSO (*N* = 3). (**D**)

*Figure 2 continued on next page*

*Figure 2 continued*

Expression of MHC protein and the fusion of myoblasts in syncytial myotubes, measured by immunofluorescence, after treatment with 15d-PGJ$_2$ (4 μM) or DMSO (*N* = 3). (**E**) Expression of MHC protein, measured by immunoblotting, in primary human skeletal myoblasts after treatment with 15d-PGJ$_2$ (1, 2, or 4 μM) or DMSO for 5 days (*N* = 3). (The Standard Deviation between replicates was plotted as error bars. Statistical significance was tested by the two-tailed Student's *t*-test ns = p > 0.05, *p < 0.05, **p < 0.01, ***p < 0.001, ****p < 0.0001.)

The online version of this article includes the following source data and figure supplement(s) for figure 2:

**Source data 1.** Uncropped and labelled gels for *Figure 2*.

**Source data 2.** Raw unedited gels for *Figure 2*.

**Figure supplement 1.** Viability of C2C12 myoblasts after treatment with 15d-PGJ$_2$ (10 μM) in differentiating medium.

on this, we treated differentiating myoblasts with 15d-PGJ$_2$ (1, 2, and 4 μM) for 5 days to measure the effects of 15d-PGJ$_2$ treatment on differentiation of myoblasts. We observed a dose-dependent decrease in the mRNA levels of *Myod1*, *Myog*, and *Myhc* in differentiating C2C12 cells after treatment with 15d-PGJ$_2$ (*Figure 2C*). There was a significant decrease in the no. of nuclei in individual MHC[+ve] fiber (~75%) in C2C12 cells treated with 15d-PGJ$_2$ (4 μM) compared to DMSO (*Figure 2D*), suggesting a decrease in the fusion of myoblasts in myotubes. We also observed a dose-dependent decrease in the protein levels of MHC in differentiating primary human myoblasts upon treatment with 15d-PGJ$_2$ (*Figure 2E*). Together, these observations suggest that 15d-PGJ$_2$ inhibits the differentiation of both mouse and human myoblasts.

## Biotinylated 15d-PGJ$_2$ covalently modifies HRas at cysteine 184

15d-PGJ$_2$ has been shown to covalently modify several proteins including p53 and NF-κB, which are involved in several key biological processes (*Marcone and Fitzgerald, 2013*). HRas was identified to be covalently modified by 15d-PGJ$_2$ at cysteine 184 in NIH3T3 and Cos1 cells (*Oliva et al., 2003*). Therefore, we tested whether 15d-PGJ$_2$ could covalently modify HRas in C2C12 cells. We treated C2C12 cells expressing the EGFP (Enhanced Green Fluorescent Protein)-tagged wild-type HRas with biotinylated 15d-PGJ$_2$ (5 μM). We then immunoprecipitated biotinylated 15d-PGJ$_2$ using streptavidin. We observed a significant increase (~3.5-fold) in the pulldown of HRas upon treatment with 15d-PGJ$_2$-Biotin compared to DMSO (*Figure 3A*), suggesting an interaction between 15d-PGJ$_2$ and HRas. To measure the role of individual C-terminal cysteines in the binding of HRas with 15d-PGJ$_2$, we treated C2C12 cells expressing the EGFP-tagged C181S and C184S mutants of HRas with biotinylated 15d-PGJ$_2$ (5 μM), and immunoprecipitated using streptavidin. We observed that the intensity of EGFP-tagged HRas was significantly decreased in cells expressing the C184S mutant (~80% decrease compared to HRas WT) but not in those expressing the C181S mutant (*Figure 3A*). This suggests that 15d-PGJ$_2$ covalently modifies HRas at cysteine 184 in C2C12 cells.

## 15d-PGJ$_2$ increases the FRET between EGFP-HRas and mCherry-RAF-RBD in wild-type and C181S mutant but not in the C184S mutant of HRas

We next tested the effect of covalent modification of HRas by 15d-PGJ$_2$ on HRas GTPase activity using FRET. mCherry-RAF-RBD is a well-characterized sensor of the activity of HRas. RAF-RBD binds to the activated HRas upon activation of HRas, allowing FRET between EGFP and mCherry (*Rocks et al., 2005*). We co-expressed EGFP-tagged HRas (EGFP-HRas) with mCherry-RAF-RBD in C2C12 myoblasts (*Figure 3B*). We measured the efficiency of FRET between EGFP and mCherry using an ImageJ plugin, FRET analyzer (*Hachet-Haas et al., 2006*). We compared the mean acceptor normalized FRET index in C2C12 myoblasts co-expressing EGFP-HRas WT and mCherry-RAF-RBD before and after treatment of 15d-PGJ$_2$ (10 μM) for 1 hr. We observed a significant increase (~30%) in the mean acceptor normalized FRET index upon treatment with 15d-PGJ$_2$ (*Figure 3C*). This suggests that 15d-PGJ$_2$ activates HRas. To measure the role of individual C-terminal cysteines in 15d-PGJ$_2$-mediated activation of HRas, we co-expressed EGFP-HRas C181S or C184S with mCherry-RAF-RBD in C2C12 myoblasts. We measured the mean acceptor normalized FRET index before and after 1 hr of treatment with 15d-PGJ$_2$ (10 μM). We observed a significant increase (~40%) in the mean acceptor normalized FRET index in cells expressing EGFP-HRas C181S upon treatment with 15d-PGJ$_2$ but not

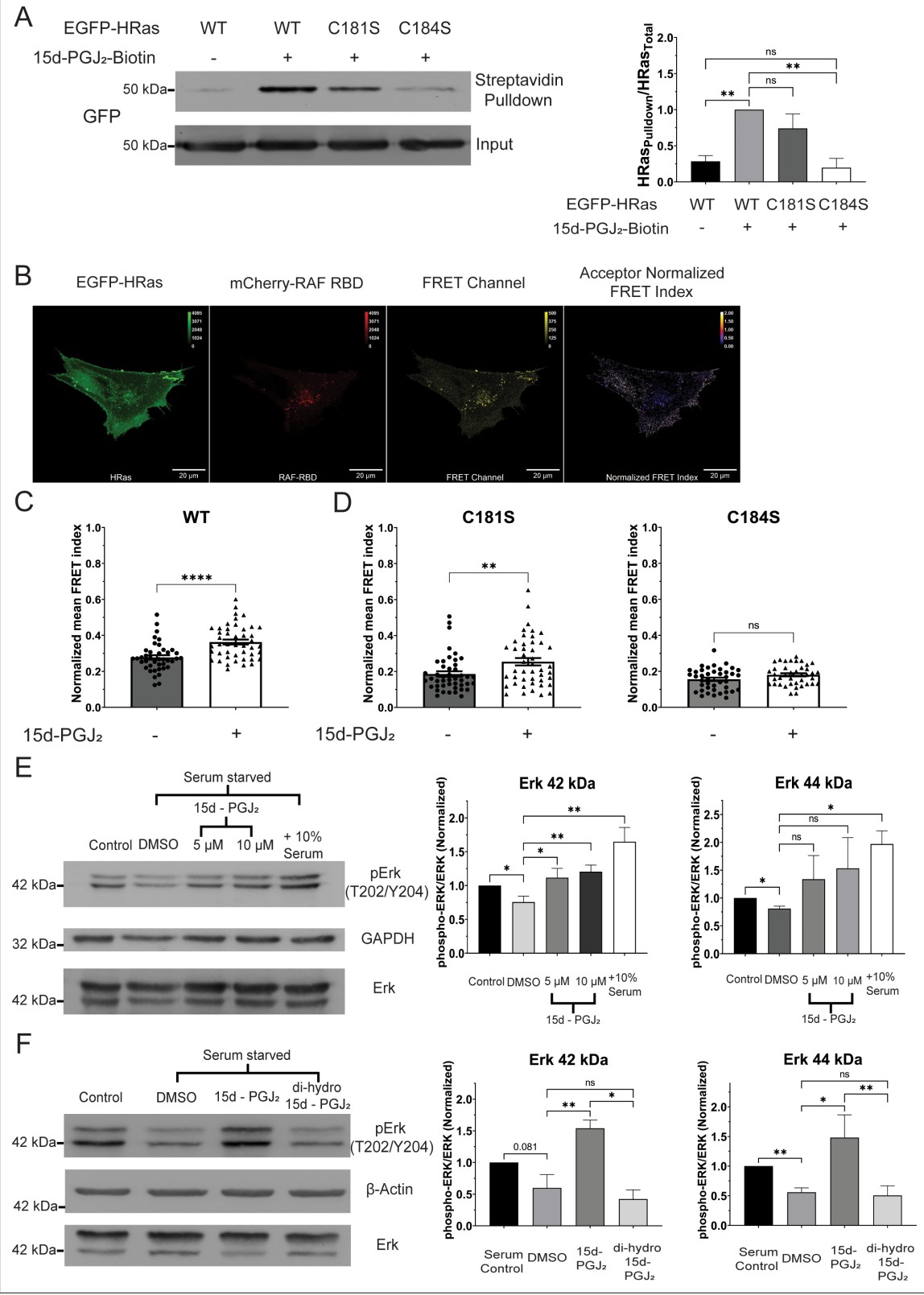

**Figure 3.** 15d-PGJ$_2$ covalently modifies HRas at cysteine 184 and activates the HRas–MAP kinase (MAPK) pathway via the electrophilic cyclopentenone ring. (**A**) Streptavidin-immunoprecipitation of EGFP-HRas, measured by immunoblotting, in C2C12 cells after 3 hr of treatment with 15d-PGJ$_2$-Biotin (5 µM) (*N* = 3). (**B**) Representative confocal micrograph of fluorescence resonance energy transfer (FRET) between EGFP-tagged HRas (EGFP-HRas) and mCherry-tagged Ras-binding domain (RBD) of RAF kinase (mCherry-RAF-RBD). (**C**) Activation of the EGFP-tagged wild-type HRas (HRas WT), measured

*Figure 3 continued on next page*

*Figure 3 continued*

by FRET, before and after 1 hr of treatment with 15d-PGJ$_2$ (10 µM) after starvation for 24 hr. (**D**) Activation of the EGFP-tagged C-terminal cysteine mutants of HRas (HRas C181S and HRas C184S), measured by FRET, before and after 1 hr of treatment with 15d-PGJ$_2$ (10 µM) after starvation for 24 hr. (**E**) Phosphorylation of Erk (42 kDa and 44 kDa), measured by immunoblotting, in C2C12 cells after 1 hr of treatment with 15d-PGJ$_2$ (5 and 10 µM) or DMSO after starvation for 24 hr ($N$ = 3). (**F**) Phosphorylation of Erk (42 and 44 kDa), measured by immunoblotting, in C2C12 cells after 1 hr of treatment with 15d-PGJ$_2$ (10 µM)/ 9,10-dihydro-15d-PGJ$_2$ (10 µM) or DMSO after starvation for 24 hr ($N$ = 3). (The Standard Deviation between replicates was plotted as error bars. Statistical significance was tested by the two-tailed Student's $t$-test ns = $p > 0.05$, *$p < 0.05$, **$p < 0.01$, ****$p < 0.0001$.)

The online version of this article includes the following source data and figure supplement(s) for figure 3:

**Source data 1.** Uncropped and labelled gels for *Figure 3*.

**Source data 2.** Raw unedited gels for *Figure 3*.

**Figure supplement 1.** Activation of HRas after treatment with 15d-PGJ$_2$.

**Figure supplement 1—source data 1.** Uncropped and labelled gels for *Figure 3—figure supplement 1*.

**Figure supplement 1—source data 2.** Raw unedited gels for *Figure 3—figure supplement 1*.

---

in cells expressing EGFP-HRas C184S (*Figure 3D*). These observations suggest that activation of HRas by 15d-PGJ$_2$ occurs in a cysteine 184-dependent manner.

## 15d-PGJ$_2$ increases phosphorylation of Erk (Thr202/Tyr204) but not Akt (S473) in C2C12 myoblasts

HRas regulates two major downstream signaling pathways, the MAP kinase (MAPK) pathway and the PI3 kinase (PI3K) pathway (*Pylayeva-Gupta et al., 2011*). We tested the effects of treatment with 15d-PGJ$_2$ on these two downstream signaling pathways by measuring the phosphorylation of Erk (42 and 44 kDa) and Akt proteins in C2C12 cells. We treated C2C12 cells with 15d-PGJ$_2$ (5 and 10 µM) or DMSO for 1 hr (after 24 hr of serum starvation) and observed a dose-dependent increase in the phosphorylation of Erk (T202/Y204) (42 kDa) but not of Erk (44 kDa) (*Figure 3E*). We did not observe an increase in the phosphorylation of Akt (S473) in C2C12 cells after treatment with 15d-PGJ$_2$ (*Figure 3—figure supplement 1C*). These observations suggest that 15d-PGJ$_2$ activates the MAPK signaling pathway, but not the PI3K signaling pathway.

15d-PGJ$_2$ contains a reactive electrophilic center in its cyclopentenone ring, that can react with cysteine residues of proteins (*Marcone and Fitzgerald, 2013*; *Shibata et al., 2002*; *Yamamoto et al., 2011*). We tested its role in activating the MAPK signaling pathway. We measured the phosphorylation of Erk (42 and 44 kDa) in C2C12 cells after treatment with cells with 9,10-dihydro-15d-PGJ$_2$ (10 µM), a 15d-PGJ$_2$ analog which is devoid of the electrophilic center, for 1 hr (after 24 hr of serum starvation). We observed that the phosphorylation of Erk (42 and 44 kDa) in C2C12 cells treated with 9,10-dihydro-15d-PGJ$_2$ was significantly reduced (~70%) as compared to the treatment with 15d-PGJ$_2$ (*Figure 3F*). This shows that 15d-PGJ$_2$ activates the HRas–MAPK signaling pathway via the electrophilic center in its cyclopentenone ring.

## 15d-PGJ$_2$ increases the localization of EGFP-tagged HRas at the plasma membrane compared to the Golgi in a C-terminal cysteine-dependent manner

15d-PGJ$_2$ covalently modifies cysteine 184 and activates HRas signaling (*Figure 3*). Reversible palmitoylation of cysteine 181 and 184 in the C-terminal tail of HRas regulates intracellular distribution and signaling of HRas. Inhibition of palmitoylation of the C-terminal cysteine 181, either by a palmitoylation inhibitor 2-bromopalmitate or by mutation to serine, causes accumulation of HRas at the Golgi compared to the plasma membrane and alters activity (*Rocks et al., 2005*). Therefore, we tested whether the modification of 15d-PGJ$_2$ alters the intracellular distribution of HRas. We co-expressed the EGFP-tagged wild type and the cysteine mutants of HRas (EGFP-HRas WT/C181S/C184S) with a previously reported marker of Golgi (*Shaner et al., 2008*) in C2C12 cells and stained the cells with plasma membrane marker WGA-633 (*Figure 4A*). We compared $R_{mean}$, the ratio of mean EGFP-HRas intensity at the Golgi to the mean HRas intensity at the plasma membrane, to measure the distribution of HRas between the plasma membrane and the Golgi. We measured the intracellular distribution of HRas between the Golgi and the plasma membrane in C2C12 cells after treatment with 15d-PGJ$_2$ (10 µM) for 24 hr in DMEM 10% serum medium and observed a significant decrease (~20%) in the $R_{mean}$

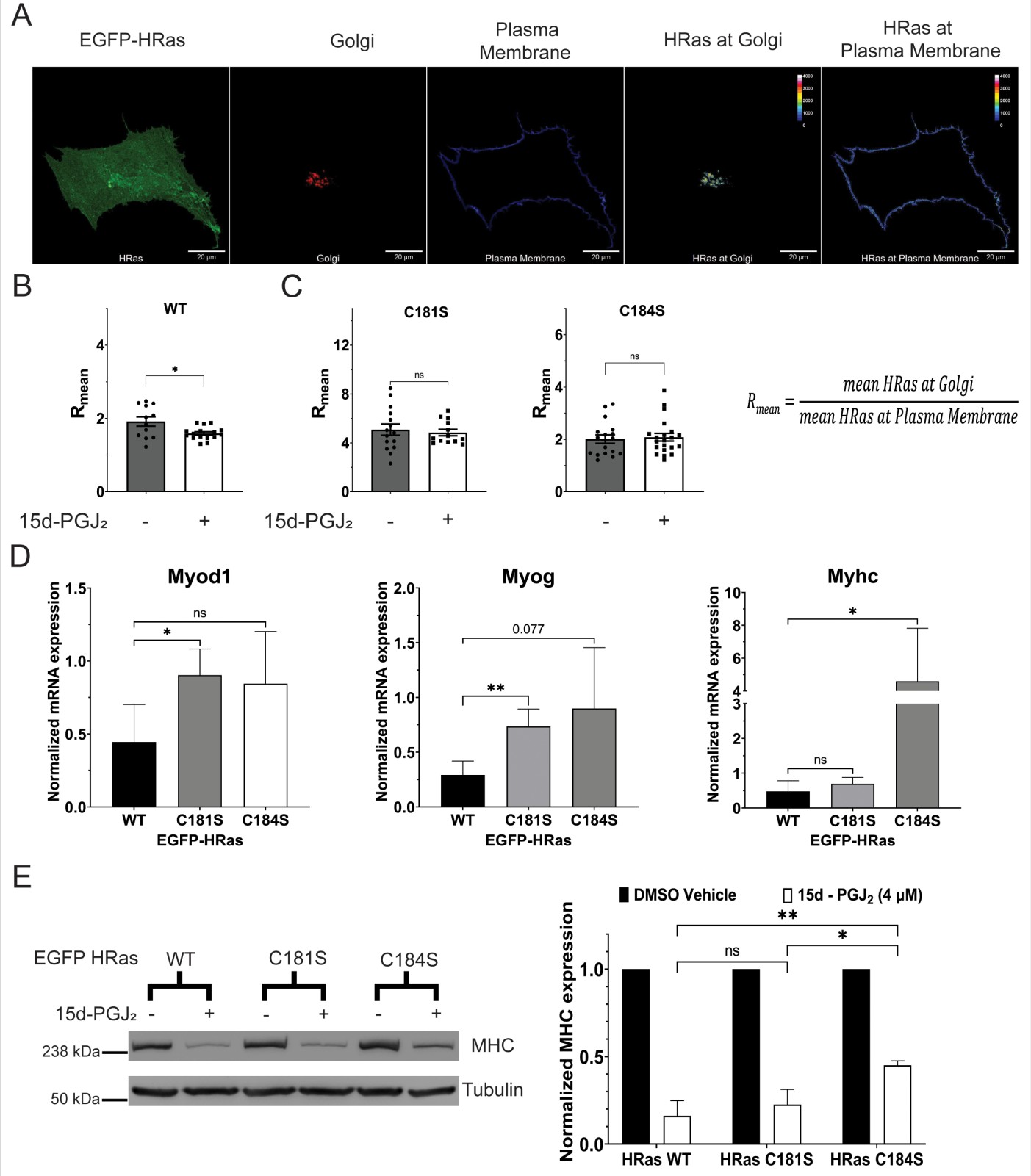

**Figure 4.** 15d-PGJ$_2$ controls the intracellular distribution of HRas and differentiation of C2C12 cells in an HRas C-terminal cysteine-dependent manner. (**A**) Representative confocal micrograph of C2C12 myoblasts showing localization of EGFP-tagged HRas between the plasma membrane (stained with Alexa Fluor 633-conjugated Wheat Germ Agglutinin) and the Golgi (labeled with TagRFP-tagged Golgi resident GalT protein). A statistic $R_{mean}$ was defined as the ratio of mean HRas intensity at the Golgi to the mean HRas intensity at the plasma membrane. (**B**) Distribution of the wild-type

*Figure 4 continued on next page*

*Figure 4 continued*

HRas between the Golgi and the plasma membrane, measured by $R_{mean}$, in C2C12 myoblasts treated with 15d-PGJ$_2$ (10 μM) or DMSO for 24 hr. (**C**) Distribution of the C-terminal cysteine mutants of HRas between the Golgi and the plasma membrane, measured by $R_{mean}$, in C2C12 myoblasts treated with 15d-PGJ$_2$ (10 μM) or DMSO for 24 hr. (**D**) Expression of mRNAs of known markers of differentiation (*Myod1*, *Myog*, and *Myhc*), measured by qPCR, in differentiating C2C12 myoblasts expressing the EGFP-tagged wild-type and the C-terminal cysteine mutants of HRas after treatment with 15d-PGJ$_2$ (4 μM) or DMSO for 5 days (*N* = 3). (**E**) Expression of MHC protein, measured by immunoblotting, in differentiating C2C12 myoblasts expressing the EGFP-tagged wild-type and the C-terminal cysteine mutants of HRas after treatment with 15d-PGJ$_2$ (4 μM) or DMSO for 5 days (*N* = 3). (The Standard Deviation between replicates was plotted as error bars. Statistical significance was tested by the two-tailed Student's *t*-test ns = p > 0.05, *p < 0.05, **p < 0.01.)

The online version of this article includes the following source data and figure supplement(s) for figure 4:

**Source data 1.** Uncropped and labelled gels for *Figure 4*.

**Source data 2.** Raw unedited gels for *Figure 4*.

**Figure supplement 1.** C-terminal cysteine-mediated intracellular distribution of constitutively active HRas (HRas V12) regulates the differentiation of C2C12 myoblasts.

**Figure supplement 1—source data 1.** Uncropped and labelled gels for *Figure 4—figure supplement 1*.

**Figure supplement 1—source data 2.** Raw unedited gels for *Figure 4—figure supplement 1*.

of C2C12 cells expressing the wild-type HRas after treatment with 15d-PGJ$_2$ (*Figure 4B*). However, we did not observe a change in the $R_{mean}$ of C2C12 cells expressing HRas C181S or HRas C184S after treatment with 15d-PGJ$_2$ (*Figure 4C*). These observations suggest that 15d-PGJ$_2$ increases the localization of HRas at the plasma membrane as compared to that in the Golgi in an HRas C-terminal cysteine-dependent manner.

## 15d-PGJ$_2$-mediated inhibition of differentiation of C2C12 cells is rescued by C181S and C184S mutants of HRas

HRas inhibits the differentiation of C2C12 myoblasts (*Engler et al., 2021*; *Konieczny et al., 1989*; *Lassar et al., 1989*; *Lee et al., 2010*; *Olson et al., 1987*; *Scholz et al., 2009*; *van der Burgt et al., 2007*). 15d-PGJ$_2$ covalently modifies cysteine 184 and activates HRas (*Figure 3*). Therefore, we tested whether the inhibition of myoblast differentiation by 15d-PGJ$_2$ depends on the activation of HRas signaling by modification of the C-terminal cysteine 184. We expressed the wild-type and the cysteine mutants of HRas (EGFP-HRas WT/C181S/C184S) in C2C12 myoblasts and treated the cells with 15d-PGJ$_2$ (4 μM) or DMSO during differentiation. We observed a decrease in the levels of mRNA of *Myhc* in C2C12 cells expressing HRas WT and HRas C181S after 5 days of treatment with 15d-PGJ$_2$. We did not observe this in expressing HRas C184S (*Figure 4D*). We also observed a significant decrease in the protein levels of MHC in differentiating C2C12 cells expressing HRas WT and HRas C181S after treatment with 15d-PGJ$_2$ (*Figure 4E*). This decrease was partially rescued in cells expressing HRas C184S (*Figure 4E*). These observations suggest that the inhibition of myoblast differentiation by 15d-PGJ$_2$ depends on modification of HRas C-terminal cysteine 184.

## Discussion

Senescence is characterized by an irreversible arrest in cell proliferation (*Hayflick, 1965*). Cells undergo senescence because of a myriad of stresses, including DNA damage, mitochondrial damage, and oncogene overexpression (*Bihani et al., 2007*; *Bihani et al., 2004*; *Casar et al., 2018*; *Chen and Ames, 1994*; *Chen et al., 1998*; *Coppé et al., 2008*; *d'Adda di Fagagna, 2008*; *d'Adda di Fagagna et al., 2003*; *Di Leonardo et al., 1994*; *Franza et al., 1986*; *Land et al., 1983*; *Robles and Adami, 1998*; *Serrano et al., 1997*; *Wiley et al., 2016b*; *Woods et al., 1997*). Senescent cells exhibit a multi-faceted physiological response, where they exhibit a flattened morphology, increase in cell size (*Chen and Ames, 1994*; *Serrano et al., 1997*), upregulation of tumor suppressor proteins (*Calabrese et al., 2009*; *Lowe et al., 2004*; *Stein et al., 1990*; *Zindy et al., 2003*), expression of neutral pH active β-galactosidase (SA β-gal) (*Dimri et al., 1995*; *Lee et al., 2006*), and altered metabolic state (*Bittles and Harper, 1984*; *Jones et al., 2005*; *Wiley and Campisi, 2021*; *Wiley and Campisi, 2016a*; *Zwerschke et al., 2003*). Arachidonic acid metabolism is upregulated in senescent cells, which leads to increased synthesis of eicosanoid prostaglandins, which regulate the physiology of senescent cells

(*Wiley et al., 2021*; *Wiley and Campisi, 2021*). Senescent cells exhibit a secretory phenotype (SASP) consisting of a variety of bioactive molecules including cytokines and chemokines, growth factors, matrix metalloproteases, etc. (*Coppé et al., 2008*). Senescent cells influence the surrounding cells via the SASP factors, which regulate proliferation, migration, and other cell biological processes in the neighboring cells (*Campisi, 2005*). SASP-mediated perturbations in the microenvironment are implicated in several senescence-associated pathologies (*Wiley and Campisi, 2021*). Senescent fibroblasts increase the proliferation of premalignant and malignant epithelial cells (*Krtolica et al., 2001*). Conditioned medium of senescent fibroblasts promoted tumorigenesis in mouse keratinocytes (*Dilley et al., 2003*). Senescent fibroblasts transform premalignant breast cancer cells into invasive, tumor-forming cells (*Parrinello et al., 2005*). Senescence in muscle stem cells induces sarcopenia via activation of the p38 MAP kinase pathway and transient inhibition of the p38 MAP kinases rejuvenates aged muscle stem cells to ameliorate sarcopenia (*Cosgrove et al., 2014*). Senescent cells inhibit the differentiation of myoblasts by secretion of IL6 by senescent muscle stem cells in myotonic dystrophy (*Conte et al., 2023*).

In this study, we show that senescent myoblasts synthesize and release eicosanoid prostaglandin 15-deoxy-$\Delta^{12,14}$-prostaglandin $J_2$ (15d-PGJ$_2$) (*Figure 1I and J*), the terminal non-enzymatic dehydration product of prostaglandin PGD$_2$ (*Shibata et al., 2002*). We used Doxo to induce senescence in C2C12 myoblasts and showed that the conditioned medium of senescent C2C12 cells inhibits differentiation of C2C12 myoblasts (*Figure 2A*). Inhibition of synthesis of PGD$_2$ by treatment of senescent cells with AT-56, a well-characterized inhibitor of PTGDS (*Hu et al., 2022*; *Hu et al., 2023a*; *Hu et al., 2023b*; *Irikura et al., 2009*), rescued this inhibitory effect of the conditioned medium on the differentiation of myoblasts (*Figure 2A*). A study has shown that prostaglandin PGD$_2$ inhibits differentiation of C2C12 myoblasts (*Veliça et al., 2010*), but the authors noted that knockout of DP1 and DP2 (the known receptors of prostaglandins PGD$_2$) does not abrogate inhibition of differentiation of myoblasts by PGD$_2$. This observation suggested that PGD$_2$ might inhibit the differentiation of myoblasts by a receptor-independent mechanism, possibly by its spontaneous non-enzymatic dehydration to 15d-PGJ$_2$. 15d-PGJ$_2$ has been suggested to be an endogenous ligand of Peroxisome proliferator-activated receptor gamma (PPARγ) (*Li et al., 2019*). However, the inhibition of PPARγ did not abrogate the inhibition of differentiation of C2C12 myoblasts by 15d-PGJ$_2$, suggesting the existence of other possible mechanisms (*Hunter et al., 2001*). 15d-PGJ$_2$ has varied effects on cell physiology in a context-dependent manner. On one hand, 15d-PGJ$_2$ promotes tumorigenesis by inducing epithelial to mesenchymal transition in breast cancer cell line MCF7 (*Choi et al., 2020*), 15d-PGJ$_2$ inhibits the proliferation of A549, H1299, and H23 lung adenocarcinoma cells via induction of Reactive Oxygen Species (ROS) and activation of apoptosis (*Slanovc et al., 2024*). Here, we show that 15d-PGJ$_2$ inhibits the proliferation and the differentiation of C2C12 myoblasts (*Figure 2B–D*).

15d-PGJ$_2$ contains an electrophilic cyclopentenone ring in its structure, allowing 15d-PGJ$_2$ to covalently modify and form Michael adducts with cysteine residues of proteins (*Shibata et al., 2002*). A previous proteomic study in endothelial cells showed biotinylated 15d-PGJ$_2$ covalently modified over 300 proteins, which regulate several physiological processes including cell cycle (MAPK1, MCM4), cell metabolism (fatty acid synthase, isocitrate dehydrogenase), apoptosis (PDCD6I), translation (elongation factor 1 and 2, EIF4A-I), intracellular transport (Importin subunit β1, Exportin 2, Kinesin 1 heavy chain) (*Marcone and Fitzgerald, 2013*). Another proteomic study in neuronal cells suggested that 15d-PGJ$_2$ modifies several proteins including chaperone HSP8A, glycolytic proteins Enolase 1 and 2, GAPDH, PKM1, cytoskeleton proteins Tubulin β2b, β actin, GFAP, etc. (*Yamamoto et al., 2011*). This study also showed modification of peptide fragments homologous to IκB kinase β, Thioredoxin, and a small molecule GTPase HRas. 15d-PGJ$_2$ has been shown to covalently modify HRas in NIH3T3 and Cos1 cells (*Oliva et al., 2003*) and IMR90 cells (*Wiley et al., 2021*). Modification by 15d-PGJ$_2$ led to the activation of HRas, judged by an increase in GTP-bound HRas. It is clear that 15d-PGJ$_2$ is capable of modifying numerous proteins in different contexts. Despite these observations, the functional relevance of these modifications in numerous contexts remains to be mapped. Here, we focused on the role of 15d-PGJ$_2$ in the context of senescence and skeletal muscle differentiation. In this study, we showed that 15d-PGJ$_2$ covalently modifies HRas at cysteine 184 but not cysteine 181 in C2C12 myoblasts (*Figure 3A*). We showed by FRET microscopy that modification of HRas by 15d-PGJ$_2$ in HRas WT and HRas C181S activates HRas in C2C12 cells, but 15d-PGJ$_2$ is unable to activate HRas C184S in this context (*Figure 3B–D* and *Figure 3—figure supplement 1A, B*). This

observation shows a direct link between the modification of HRas by 15d-PGJ$_2$ and the activation of HRas GTPase.

HRas activates two major downstream signaling pathways, the HRas–MAPK and the HRas–PI3K pathway (*Pylayeva-Gupta et al., 2011*). We showed that covalent modification of HRas by 15d-PGJ$_2$ via the electrophilic cyclopentenone ring activates HRas (*Figure 3C, D*) and activates the HRas–MAPK pathway, demonstrated by an increase in the phosphorylation of Erk after treatment with 15d-PGJ$_2$ (*Figure 3E, F*). However, we did not observe activation of the HRas–PI3K pathway, as we did not see an increase in the phosphorylation of Akt after treatment with 15d-PGJ$_2$ (*Figure 3—figure supplement 1C*). MAPK and PI3K pathways are known regulators of muscle differentiation (*Bennett and Tonks, 1997*; *Rommel et al., 1999*), where inhibition of the RAF–Mitogen-Activated Protein Kinase Kinase (MEK)–Erk pathway or activation of the PI3K pathway promotes the differentiation of myoblasts. Preferential activation of the HRas–MAPK pathway over the HRas–PI3K pathway after treatment with 15d-PGJ$_2$ can be a possible mechanism by which 15d-PGJ$_2$ can inhibit the differentiation of myoblasts. HRas is known to regulate the differentiation of myoblasts in different contexts. Constitutively active HRas signaling by expression of oncogenic HRas mutant (HRas V12) leads to inhibition of differentiation of myoblasts (*Konieczny et al., 1989*; *Lassar et al., 1989*; *Olson et al., 1987*; *van der Burgt et al., 2007*). Here, we showed that the inhibition of differentiation of myoblasts after 15d-PGJ$_2$ is partially rescued in cells expressing the C184S mutant of HRas but not the wild type or the C181S mutant (*Figure 4D, E* and *Figure 4—figure supplement 1E*). HRas C184S did not get modified by 15d-PGJ$_2$ (*Figure 3A*). These observations suggest that the inhibition of differentiation of myoblasts by 15d-PGJ$_2$ is partially dependent on the covalent modification of HRas by 15d-PGJ$_2$.

Cysteine 181 and 184 in the C-terminal of HRas regulate the intracellular distribution of HRas between the plasma membrane and the Golgi by reversible palmitoylation and de-palmitoylation (*Rocks et al., 2005*). Inhibition of the palmitoylation of C-terminal cysteine 181, either by treatment with protein palmitoylation inhibitor 2-bromopalmitate or mutation of cysteine to serine, leads to accumulation of HRas at the Golgi. Intracellular localization of HRas maintains two distinct pools of HRas activity, where the plasma membrane pool shows a faster activation followed by short kinetics and the Golgi pool shows a slower activation but a sustained activation (*Agudo-Ibáñez et al., 2015*; *Busquets-Hernández and Triola, 2021*; *Lorentzen et al., 2010*; *Rocks et al., 2005*). We showed that the covalent modification of HRas by 15d-PGJ$_2$ alters the intracellular distribution of HRas. We showed that the covalent modification of HRas by 15d-PGJ$_2$ leads to an increase in the localization of the wild-type HRas at the plasma membrane compared to the Golgi (*Figure 4B*). We did not observe any changes in the intracellular distribution of HRas C181S or HRas C184S after treatment with 15d-PGJ$_2$ (*Figure 4C*). HRas C184S is not modified by 15d-PGJ$_2$, but HRas C181S is modified by 15d-PGJ$_2$ (*Figure 3A*). This suggests that the intracellular redistribution of HRas due to covalent modification by 15d-PGJ$_2$ at cysteine 184 requires palmitoylation of cysteine 181.

Previous reports suggest that downstream signaling of HRas depends on the intracellular localization of HRas (*Rocks et al., 2005*; *Santra et al., 2019*). For example, targeted localization of HRas at the Endoplasmic Reticulum (ER) membrane-induced expression of the cell migration genes. Localization of HRas at the plasma membrane showed a strong correlation with the expression of cell cycle genes, particularly the MAPK signaling pathway. Localization of HRas at the plasma membrane also showed a negative correlation with genes associated with the PI3K–Akt pathway. Here, we showed that the intracellular distribution of HRas regulates differentiation of myoblasts. In order to show this, we used the constitutively active mutant of HRas (HRas V12) which has been shown to inhibit the differentiation of myoblasts (*Engler et al., 2021*; *Konieczny et al., 1989*; *Lassar et al., 1989*; *Olson et al., 1987*; *Scholz et al., 2009*; *van der Burgt et al., 2007*). We expressed cysteine mutants of HRas V12 in C2C12 myoblasts and found that HRas V12 C181S localized predominantly at the Golgi whereas HRas V12 and HRas V12 C184S localized at both the plasma membrane and the Golgi (*Figure 4—figure supplement 1A*). When differentiated, we observed that C2C12 cells expressing HRas V12 C181S differentiated but HRas V12 or HRas V12 C184S did not differentiate (*Figure 4—figure supplement 1B–D*). These observations suggest alteration of intracellular distribution of HRas affects the HRas-mediated inhibition of the differentiation of myoblasts.

Doxo is a widely used chemotherapy agent for the treatment of cancers (*Johnson-Arbor and Dubey, 2022*). Treatment with Doxo induces senescence. Doxo-mediated DNA damage leads to p53-, p16-, and p21-dependent senescence in human fibroblasts (*Di Leonardo et al., 1994*; *Robles*

*and Adami, 1998*). On the other hand, treatment with Doxo leads to a decrease in muscle mass and cross-sectional area, leading to chemotherapy-induced cachexia (*Hiensch et al., 2020*). Several mechanisms have been proposed behind chemotherapy-induced cachexia, including the generation of reactive oxygen species (*Gilliam and St Clair, 2011*), activation of proteases like calpain and caspases (*Gilliam et al., 2012*; *Smuder et al., 2011*), and impaired insulin signaling (*de Lima Junior et al., 2016*). This study provides a possible mechanism behind chemotherapy-induced loss of muscle mass and functioning. Induction of senescence in myoblasts by treatment with Doxo could lead to increased synthesis and release of 15d-PGJ$_2$ by senescent cells which could be taken up by myoblasts in the microenvironment. The lipid could covalently modify and activate HRas at cysteine 184 to inhibit the differentiation of myoblasts. Therefore, targeting the synthesis and release of 15d-PGJ$_2$ by senescent cells could serve as an important target to promote skeletal muscle homeostasis in cancer patients.

## Materials and methods

### Plasmids

Unmutated and cysteine mutants of HRas WT [HRas WT, HRas-C181S, and HRas-C184S] and HRas V12 [HRas V12, HRas V12-C181S, HRas V12-C184S] were cloned in the pEGFPC1 vector (Clontech) by restriction digestion–ligation method. Constructs of wild-type HRas were PCR amplified from a previously available HRas construct in the lab with construct-specific primers. Proper nucleotide additions were made to the forward primer to maintain the EGFP ORF, marking a 7 amino acid linker between the proteins. The construct sequences were confirmed by Sanger sequencing. GalT-TagRFP construct was a gift from Prof. Satyajit Mayor and was used to mark the Golgi. mCherry-RAF-RBD construct was a gift from Prof. Phillipe Bastiens and was used to measure the activity of HRas GTPase using FRET.

### Cell maintenance

C2C12 mouse myoblasts (CRL-1772) and MCF7 human breast adenocarcinoma cells (HTB-22) were obtained from ATCC and were maintained in DMEM complete medium at 37°C, 5% CO$_2$. For experiments, the cells were trypsinized with 0.125% trypsin–Ethylenediaminetetraacetic acid (EDTA) (Gibco) and were seeded in required numbers in cell culture dishes. Human Skeletal Muscle Myoblast (CC-2580) were obtained from Lonza and were maintained in DMEM Skeletal Muscle growth medium at 37°C, 5% CO$_2$. For experiments, the cells were trypsinized with 0.125% trypsin–EDTA (Gibco) and were seeded in required numbers in cell culture dishes. All cultures tested negative for mycoplasma checked by Mycoalert Mycoplasma Detection Kit (Lonza).

### Conditioned media collection

C2C12 cells seeded in 60 mm dishes were treated with Doxo (150 nM) for 3 days. The media was then changed to DMEM complete medium without Doxo for 19 days after treatment with Doxo. The cells were treated with DMSO or AT-56 (30 µM) in the DMEM complete medium for 2 days. On Day 21, the cells were treated with DMSO or At-56 in DMEM Starvation medium for 3 days. The media was then collected and centrifuged at 1000 × *g*, Room Temperature for 5 min. The media was then stored at −80°C after flash freezing in liq. N$_2$ till further requirement.

### Treatments

15d-PGJ$_2$ (Cayman Chemical Company) dissolved in DMSO (10 mM) was diluted in DMEM media for experiments. 9,10-dihydro-15d-PGJ$_2$ (Cayman Chemical Company) dissolved in DMSO (10 mM) was appropriately diluted in DMEM media for experiments. DMSO was used as vehicle control. A media change of the same composition was given every 24 hr. C2C12 and MCF7 cells with 70–80% confluency were treated with Doxo for 3 days. After 72 hr, Doxo was removed from the medium and the cells were kept for 10 more days with media change every 3 days till the end of the experiment. C2C12 cells transfected with EGFP-HRas WT/C181S/C184S in 35 mm dishes were treated with 15d-PGJ$_2$-Biotin (5 µM) in DMEM Hi Glucose medium (Gibco) supplemented with 1% penicillin–streptomycin–glutamine (Gibco) without fetal bovine serum for 3 hr. Conditioned medium collected from senescent cells was thawed at 37°C. The medium was then supplemented with 2% heat-inactivated horse serum and 1% penicillin–streptavidin–glutamine. C2C12 myoblasts were treated with the conditioned medium and were given a media change every 48 hr.

## Transfections

C2C12 cells were seeded in 35 mm dishes to achieve confluency of ~60–70%. For western blot, immunoprecipitation, and differentiation experiments, the cells were transfected with EGFP-tagged HRas WT/HRas-C181S/HRas-C184S/HRas V12/HRas V12-C181S/HRas V12-C184S using the jetPRIME transfection reagent (Polyplus) using the manufacturer's protocol. For measuring the intracellular distribution of HRas, the cells were reverse transfected with EGFP-tagged HRas WT/HRas-C181S/HRas-C184S/HRas V12/HRas V12-C181S/HRas V12-C184S and GalT-TagRFP, a Golgi apparatus marker protein tagged with red fluorescent TagRFP protein using the jetPRIME transfection reagent using the manufacturer's protocol. For measuring the activity of HRas, the cells seeded in imaging dishes (iBidi) were transfected with EGFP-HRas and mCherry-RAF-RBD using jetPRIME transfection reagent (Polyplus) using the manufacturer's protocol. Transfection efficiency was confirmed by checking for Green Fluorescence Protein (GFP) and Red Fluorescence Protein (RFP) fluorescence after 24 hr of transfection.

## Myoblast differentiation

C2C12 cells were treated with either 15d-PGJ$_2$ or DMSO in the C2C12 differentiation medium. The cells were given a media change of the same composition every 24 hr. The cells were harvested after 5 days of 15d-PGJ$_2$ treatment for either RNA or protein isolation. Human Skeletal Muscle Myoblast cells were treated with DMSO or 15d-PGJ$_2$ in the Skeletal Muscle Differentiation medium. A media change of the same composition was given every 24 hr. The cells were harvested after 5 days of treatment for protein isolation.

## X-Gal staining

Proliferative and Doxo-treated C2C12 cells were fixed with 0.25% glutaraldehyde, washed with Phosphate Buffer Saline (PBS), and incubated overnight in X-gal staining solution at 37°C in a CO$_2$-free chamber. The presence of the Indigo blue product was confirmed using the Ti2 widefield inverted microscope (Nikon).

## Immunoprecipitation (IP)

C2C12 cells transfected with EGFP-HRas and treated with 15d-PGJ$_2$-Biotin were harvested and lysed in Radioimmunoassay buffer supplemented with protease and phophatase inhibitors (RIPA-PP buffer) and the lysate was centrifuged at 15,000 rpm, 4°C, 30 min. Protein estimation was done using the BCA assay kit (G Biosciences). 100 µg of protein was loaded on 10 µl MyOne Streptavidin C1 dynabeads blocked with 1% Bovine Serum Albumin (BSA) in IP washing buffer. The lysate–streptavidin mix was incubated at 4°C, 10 rpm overnight. The beads were then washed with IP washing buffer and then boiled in 20 µl Laemmlli buffer. 15 µl of the beads were loaded on 12% Sodium Dodecyl Sulphate (SDS)–polyacrylamide gel for detection of EGFP-HRas by immunoblotting using EGFP antibody.

## Western blotting

For measuring Erk/Akt phosphorylation in C2C12 cells were seeded in 35 mm dishes. 1 × 35 mm dish was harvested in RIPA-PP the next day, while the rest were incubated in DMEM starvation medium at 37°C. The cells were treated with 15d-PGJ$_2$ after 24 hr of starvation at 37°C. The cells were harvested at 1 hr after treatment in RIPA-PP. Protein quantification was done using BCA assay (G Biosciences) using the manufacturer's protocol. An equal mass of proteins was loaded onto a 12% SDS–polyacrylamide gel in Laemmlli buffer. The proteins were transferred onto a Polyvinylidene difluoride (PVDF) membrane and were probed with phospho-Erk/Erk antibodies for measuring Erk phosphorylation and with phospho-Akt/Akt antibodies for measuring Akt phosphorylation. For measuring the expression of Myosin Heavy Chain, C2C12 cells expressing EGFP-tagged HRas WT/HRas-C181S/HRas-C184S/HRas V12/HRas V12-C181S/HRas V12-C184S or Human Skeletal Muscle Myoblasts were seeded in 35 mm dishes and were harvested in RIPA-PP after 5 days of differentiation. Protein quantification was done using BCA assay (G Biosciences) using the manufacturer's protocol. An equal mass of proteins was loaded onto an 8% SDS–polyacrylamide gel in Laemmlli buffer. The proteins were transferred onto a PVDF membrane and were probed with Myosin Heavy Chain Antibody.

## qPCR

C2C12 cells, untransfected or expressing EGFP-tagged HRas WT/HRas-C181S/HRas-C184S and treated with DMSO/15d-PGJ$_2$, or expressing EGFP-tagged HRas V12/HRas V12-C181S/HRas

V12-C184S were lysed in TRIZol at the end of the experiment (Invitrogen). RNA was isolated from the lysate by the chloroform–isopropanol method using the manufacturer's protocol. The RNA was quantified and 1.5 µg of RNA was used to prepare cDNA using PrimeScript 1st strand cDNA Synthesis Kit (Takara Bio) and random hexamer primer. Gene expression for differentiation markers was measured by qPCR using PowerUp SYBR Green Master Mix (Applied Biosystems) and previously reported qPCR primers (*Supplementary file 1a*; *Wang et al., 2012*). Relative gene expression was quantified using the $\Delta\Delta C_T$ method (*Livak and Schmittgen, 2001*) with 18s rRNA as an internal loading control and DMSO vehicle as an experimental control. For animal studies, $\Delta C_T$ values were plotted to show the expression of the mRNAs of the genes of interest to the expression of 18s rRNA in animals treated with Saline or Doxo.

## Immunofluorescence

C2C12 cells were seeded in 35 mm dishes (Corning) on glass coverslips (Blue Star) coated with 0.2% Gelatin (Porcine, Sigma-Aldrich) and were fixed with the fixative solution at the end of the experiment. The cells were then permeabilized and blocked with the blocking solution and were then incubated with Myosin Heavy Chain antibody in the blocking solution overnight. The cells were then washed with 1× PBS, incubated with fluorophore tagged secondary antibody, and were mounted in Prolong gold antifade medium with 4',6-diamidino-2-phenylindole (DAPI) (Invitrogen). The cells were then imaged under the FV3000 inverted confocal laser scanning microscope (Olympus-Evident) using appropriate lasers and detectors.

## Confocal microscopy for measuring HRas distribution between the Golgi and the plasma membrane

C2C12 cells expressing EGFP-tagged HRas WT/HRas-C181S/HRas-C184S+GalT-TagRFP were starved overnight in DMEM starvation medium and treated with DMSO or 15d-PGJ$_2$ (10 µM) in DMEM complete medium for 24 hr, with a medium change at 12 hr post-treatment. The cells were then fixed with the fixative solution at RT, washed with PBS, and stained for plasma membrane with Alexa Fluor 633-conjugated Wheat Germ Agglutinin (WGA-633) (Invitrogen). The cells were washed with PBS and were then mounted on glass slides in ProLong Gold Antifade Mounting medium (Invitrogen). C2C12 cells expressing EGFP-tagged HRas V12/HRas V12-C181S/HRas V12-C184S were also fixed with the fixative solution at RT, washed with PBS, stained with WGA-633, and mounted on slides in Prolong gold antifade medium (Invitrogen). The cells were imaged with the FV3000 inverted confocal laser scanning microscope (Olympus-Evident) using appropriate lasers and detectors. Preliminary image processing was done using ImageJ (NIH), while batch analysis of HRas at the plasma membrane and the Golgi complex was done using a custom MATLAB script (see 'Source Code File 1'), where EGFP-HRas image was overlayed onto the GalT-TagRFP and WGA-633 image to obtain HRas localization at the Golgi complex and the plasma membrane, respectively. A ratio of mean HRas intensity at the Golgi complex to that of at the plasma membrane ($R_{mean}$) was calculated and was used to compare HRas distribution between treatments.

## FRET confocal microscopy to measure the intracellular activity of HRas

C2C12 cells expressing EGFP-tagged HRas WT/HRas C181S/HRas C184S and mCherry-RAF-RBD were starved overnight in the DMEM starvation medium. The cells were imaged with the FV3000 inverted confocal laser scanning microscope (Olympus-Evident) using the following lasers and detectors:

1. Donor channel: 488 nm excitation, 510 (+/−) 20 nm detection.
2. Acceptor channel: 561 nm excitation, 630 (+/−) 50 nm detection.
3. FRET channel: 488 nm excitation, 630 (+/−) 50 nm detection.

The cells were then treated with 15d-PGJ$_2$ (10 µM) for 1 hr and were imaged using the same imaging parameters. C2C12 cells expressing EGFP-HRas or mCherry-RAF-RBD only were used to calculate the bleed-through corrections (EGFP emission at 630 (+/−) 50 nm, and Excitation of mCherry by 488 nm laser). Preliminary processing was done using ImageJ (NIH). The FRET index was calculated using the FRET and co-localization analyzer plugin (*Hachet-Haas et al., 2006*). The FRET index was then divided by the intensity of the Acceptor channel to normalize the variation in the expression of

mCherry. We used the mean normalized FRET index to compare the activity of HRas before and after treatment with 15d-PGJ$_2$.

## Quantification of myotube fusion index

Differentiated C2C12 myoblasts were immunostained for MHC and DAPI and were imaged on the FV3000 inverted confocal laser scanning microscope (Olympus-Evident). Analysis of the fusion index was done using the Myotube Analyzer Software (*Noë et al., 2022*). DAPI and MHC images were thresholded to remove background noise. The images were converted to binary masks and the channels were overlayed to obtain the no. of nuclei overlaying with MHC[+ve] fibers. The fusion index was calculated as the percentage ratio of no. of nuclei overlaying the MHC[+ve] fibers to the total no. of nuclei in the field of view.

## Quantification of cell doubling time

Cells were counted every 24 hr and the normalization was done to the number of cells counted on day 0 of the treatment (to consider attaching efficiency and other cell culture parameters). Doubling time was calculated as the reciprocal of the slope of the graph of $\log_2$(normalized cell number) vs time.

## MTT assay

An equal number of C2C12 cells were seeded in 96 well plates in replicates. MTT assay was done at the end of the experiment using the manufacturer's protocol. MTT reagent (Sigma-Aldrich) was dissolved in 1× Dulbecco's modified Phosphate Buffer Saline (DPBS) (5 mg/ml) and was filter sterilized. MTT reagent was added to each well and the cells were incubated at 37°C, 5% $CO_2$ for 3 hr. The medium was removed at the end of the incubation and the precipitated crystals were dissolved in DMSO at 37°C, 5% $CO_2$ for 15 min. Absorbance at 570 nm was recorded using the varioskan multimode plate reader (Thermo Scientific).

## Animal experiments

Mice were maintained at BLiSC Animal Care and Resource Centre (ACRC). All the procedures performed were approved by the Internal Animal Users Committee (IAUC) and the Institutional Animal Ethics Committee (IAEC). 12- to 15-week-old C57BL/6J (JAX#000664) mice were injected intraperitoneally with 5 mg/kg Doxo four times, once every 3 days. Intraperitoneal injection of Saline was used as a control. The mice were sacrificed on Day 11 after the first injection. Hindlimb muscles from four animals (control and treated with Doxo each) were used for qPCR analysis and Hindlimb muscles from three animals (control and treated with Doxo each) were used for immunohistochemical analysis.

## Lipid extraction and detection of 15d-PGJ$_2$ by mass spectrometry

For lipid extraction, 675 µl of methanol with 3% Formic Acid was added to the 1.35 ml of conditioned medium, making a uniform sample volume of 2.025 ml. Subsequently, 1 ml of ethyl acetate was added to each sample and mixed vigorously. Phase separation was done by centrifuging the mixture (12,000 × *g*, 4°C for 10 min), and the organic phase containing the lipid was collected. This process was repeated thrice in total and all the organic phases were combined and dried under a nitrogen stream at RT. The residues were resuspended in 100 µl of 50% acetonitrile in water with 0.1% FA and were subjected to mass spec analysis using the Waters Acquity UPLC class I system The detection of 15d-PGJ$_2$ was performed using an electrospray ionization source operating in the negative ion mode and a quadrupole trap mass spectrometer (AB SCIEX QTRAP 5500) connected to a Waters Acquity UPLC class I system (Waters, Germany) outfitted with a binary solvent delivery system with an online degasser and a column manager with a column oven coupled to a Ultra Performance Liquid Chromatography (UPLC) autosampler. 5 µl samples were injected into the union for analysis. Solvent A consisted of 0.1% ammonium acetate in water and solvent B was 0.1% ammonium acetate in a mixture of acetonitrile/water (95:5). For each run, the Liquid Chromatography (LC) gradient was: 0 min, 20% B; 0.5 min, 20% B; 1.5 min, 90% B; 2.5 min, 20% B; 3 min, 20% B. Analyte detection was performed using multiple reaction monitoring, 315.100 ➔ 271.100 and 315.100 ➔ 203.100. Source parameters were set as follows: capillary voltage 3.8 kV, desolvation gas flow 25 l/hr, source temperature 350°C, ion source gas 1 flow 40 l/hr, and ion source gas 2 flow 40 l/hr. Acquisition and quantification were completed with Analyst 1.6.3 and Multiquant 3.0.3, respectively

(method adopted from *Morgenstern et al., 2018*). For the standards, 1.35 ml media of different known concentrations of 15d-PGJ$_2$ (100, 200, 300, 400, and 500 nM) were prepared and subjected to the same extraction procedure as that of the conditioned medium. A standard curve was plotted with the known concentration and the mass spec peak area, and the concentration of the lipid in samples was calculated.

## Reagents

- DMEM complete medium: DMEM Hi Glucose medium (Gibco) supplemented with 1% penicillin–streptomycin–glutamine (Gibco) and heat-inactivated 10% fetal bovine serum (US origin) (Gibco).
- Basal conditioned medium: DMEM Hi Glucose medium (Gibco) supplemented with 1% penicillin–streptomycin–glutamine (Gibco) and heat-inactivated 2% fetal bovine serum (US origin) (Gibco).
- C2C12 differentiation medium: DMEM Hi Glucose medium (Gibco) supplemented with 2% horse serum (Gibco) and 1% penicillin–streptomycin–glutamine (Gibco).
- DMEM starvation medium: DMEM Hi Glucose medium (Gibco) supplemented with 0.2% heat-inactivated fetal bovine serum (US origin) (Gibco) and 1% penicillin–streptomycin–glutamine (Gibco).
- RIPA-PP buffer: RIPA buffer (Invitrogen) supplemented with protease inhibitor cocktail (Roche) and 5 mM sodium fluoride and 5 mM sodium orthovanadate.
- TBS-T buffer: 50 mM Tris–Cl (pH = 7.5), 150 mM NaCl and 0.1% Tween-20 in water.
- PBS: 2.67 mM KCl, 1.47 mM KH$_2$PO$_4$, 137.93 mM NaCl, 8.06 mM Na$_2$HPO$_4$ in water.
- IP washing buffer: 150 mM NaCl, 0.1% SDS, 1% NP-40 in 50 mM Tris–Cl (pH = 7).
- Fixative solution: 4% (wt/vol) paraformaldehyde (Sigma-Aldrich) in PBS.
- Blocking solution: 2% heat-inactivated fetal bovine serum, 0.2% BSA, 0.2% Triton-X, 0.05% NaN$_3$ in PBS.
- Skeletal muscle growth medium: DMEM low glucose medium (Gibco), supplemented with 1% penicillin–streptomycin–glutamine (Gibco), heat-inactivated 10% fetal bovine serum (US origin) (Gibco), bovine fetuin (50 µg/ml) (Sigma-Aldrich), dexamethasone (0.4 µg/ml), and human epidermal growth factor (hEGF) (10 ng/ml).
- Skeletal muscle differentiation medium: DMEM low glucose medium (Gibco) supplemented with 2% horse serum, 1% penicillin–streptomycin (Gibco), and 1% N2 supplement.

## Acknowledgements

We thank Prof. Satyajit Mayor (NCBS), Prof. Phillipe Bastiens, and Prof. Apurva Sarin (InStem) for providing the wild-type HRas construct, the mCherry-RAF-RBD construct, and the vector backbones, respectively. We thank Dr. Neetu Saini (InStem) for her help with setting up the cell culture facility. We thank Dr. Kamlesh Kumar Yadav and Ms. Sudeshna Saha for their help during the project. We thank the Central Imaging and Flow Cytometry Facility (CIFF) (NCBS-InStem) for their support with microscopy. We thank the Animal Care and Resource Centre (ACRC) (NCBS-InStem) for supporting mouse experiments. We thank the Mass Spectrometry facility (NCBS-InStem) for supporting the mass spectrometry work.

## Additional information

### Funding

| Funder | Grant reference number | Author |
| --- | --- | --- |
| Science and Engineering Research Board | SUPRA | Arvind Ramanathan |

The funders had no role in study design, data collection, and interpretation, or the decision to submit the work for publication.

## Author contributions
Swarang Sachin Pundlik, Conceptualization, Data curation, Formal analysis, Supervision, Validation, Visualization, Methodology, Writing - original draft, Writing – review and editing; Alok Barik, Snehasudha Subhadarshini Sahoo, Data curation, Methodology, Writing – review and editing; Ashwin Venkateshvaran, Software, Formal analysis, Validation, Methodology, Writing – review and editing; Mahapatra Anshuman Jaysingh, Software, Methodology; Raviswamy GH Math, Data curation, Software, Methodology; Heera Lal, Methodology; Maroof Athar Hashmi, Formal analysis, Validation; Arvind Ramanathan, Conceptualization, Resources, Data curation, Supervision, Funding acquisition, Investigation, Visualization, Methodology, Writing - original draft, Project administration, Writing – review and editing

## Author ORCIDs
Swarang Sachin Pundlik ⬥ https://orcid.org/0000-0003-4024-5160
Arvind Ramanathan ⬥ https://orcid.org/0000-0002-0375-3740

## Ethics
All the animals were handled and the procedures (injection, sacrifice) were performed according to protocols approved by the Internal Animal Users Committee (IAUC) and the Institutional Animal Ethics Committee (IAEC) of NCBS-InStem (Permit Number: INS-IAE-2023/03(N)).

Reviewer #1 (Public review): https://doi.org/10.7554/eLife.95229.3.sa1
Reviewer #2 (Public review): https://doi.org/10.7554/eLife.95229.3.sa2
Author response https://doi.org/10.7554/eLife.95229.3.sa3

---

# Additional files

## Supplementary files
• Supplementary file 1. List of resources used. (A) List of primers used for qPCR. (B) List of reagents and the catalog numbers. (C) List of antibodies used for the western blot, immunoprecipitation, and immunofluorescence.
• MDAR checklist
• Source code 1. MATLAB script (.m file) used to calculate Rmean.

## Data availability
The code used to compute the distribution of HRas between the plasma membrane and the Golgi is provided as *Source code 1*.

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
