## [Editor Report · eLife assessment]

This manuscript outlines an interaction between senescence-related 15d-PGJ2 and the proliferation and differentiation of myoblasts, with potential implications for muscle health. This manuscript is **useful** in understanding the role of lipid metabolite 15d-PGJ2 in myoblast proliferation and differentiation. However, in its current form, the manuscript is **incomplete** as there are several concerns in the statistical analysis, lack of clarity on the mechanistic details, and concerns about the use of an immortalized C2C12 myoblasts cell line to draw major conclusions related to senescence-associated secreted phenotype.

---

## [Referee Report · Reviewer #1 (Public review)]

Summary:

The authors show that upon treatment with Doxorubicin (Doxo), there is an increase in senescence and inflammatory markers in the muscles. They also show these genes get upregulated in C2C12 myoblasts when treated with conditioned media or 15d-PGJ2. 15dPGJ2 induces cell death in the myoblasts, decreases proliferation (measured by cell numbers), and decreases differentiation and fusion. 15d-PGJ2 modified Cys184 of HRas, which is required for its activation as indicated by the FRET analysis with RAF RBD. They also showed that 15d-PGJ2 activates ERK signaling, but not Akt signaling, through the electrophilic center. 15d-PGJ2 inhibits Golgi localization of HRAS (only WT, not C181 or C184 mutant). They also showed that expressing the WT HRas followed by 15d-PGJ2 treatment led to a decrease in the levels of MHC mRNA and protein, and this defect is dependent on C184. This is a well-written manuscript with interesting insights into the mechanism of action of 15d-PGJ2. However, some clarification and experiments will help the paper advance the field significantly.

Strengths:

The data clearly shows that 15d-PGJ2 has a negative role in the myoblast cells and that it leads to modification of HRas protein. Moreover, the induction of biosynthetic enzymes in the PGD2 pathway also supports the induction of 15d-PGJ2 in Doxorubicin-treated cells. Both conditioned media experiments and the 15d-PGJ2 experiments show that 15d-PGJ2 could be the active component secreted by the senescent myoblasts.

Weaknesses:

The genes that are upregulated in the muscles upon injection with Doxo are also markers for inflammation. Since Doxo is also known to induce systemic inflammation, it is important to delineate these two effects (Inflammatory cells vs senescent cells). The expression of beta Gal and other markers of senescence in the tissue sections will help to delineate these.

In Figure 2, where the defect in the differentiation of myoblasts upon treatment with 15d-PGJ2 is shown, most of the cells die within 48 hours at higher concentrations, making it difficult to perform the experiments. This also shows that 15d-PGJ2 was toxic to these cells. Lower concentrations show a decrease in the differentiation based on the lower number of nuclei in fibers and low expression of MyoD, MyoG, and MHC. However, it is unclear if this is due to increased cell death or defective differentiation. It would be a lot more informative if the cell count, cell division, and cell death could be plotted for these concentrations of the drug during the experiment. Also, in the myoblast experiments, are the effects of treatment with Dox reversible?

In Figure 3, most of the experiments are done at a high concentration, which induces almost complete cell death within 48 hours. Even at such a high concentration of 15dPGJ2, the increase in ERK phosphorylation is minimal.

The experiment Figure 4C shows that C181 and C84 mutants of the HRas show higher levels in Golgi compared with WT. However, this could very well be due to the defect in palmitoylation rather than the modification with 15d-PGJ2. Though the authors allude to the possibility that intracellular redistribution of HRas by 15d-PGJ2 requires C181 palmitoylation, the direct influence of C184 modification on C181 palmitoylation is not shown. To have a meaningful conclusion, the authors need to compare the palmitoylation and modification with 15d-PGJ2.

To test if the inhibition of myoblast differentiation depends on HRas, they overexpressed the HRas and mutants in the C2C12 lines. However, this experiment does not take the endogenous HRAs into consideration, especially when interpreting the C184 mutant. An appropriate experiment to test this would be to knock down or knock out HRas (or make knock-in mutations of C184) and show that the effect of 15d-PGJ2 disappears. Moreover, in this specific experiment, it is difficult to interpret without a control with no HRas construct and another without the 15d-PGJ2 treatment.

Moreover, the overall study does not delineate the toxic effects of 15d-PGJ2 from its effect on the differentiation.

---

## [Referee Report · Reviewer #2 (Public review)]

Summary:

In this study, Swarang and colleagues identified the lipid metabolite 15d-PGJ2 as a potential component of senescent myoblasts. They proposed that 15d-PGJ2 inhibits myoblast proliferation and differentiation by binding and regulating HRas, suggesting its potential as a target for restoring muscle homeostasis post-chemotherapy.

Strengths:

The regulation of HRas by 15d-PGJ2 is well controlled.

Weaknesses:

(1) I still think the novelty is limited by previous published findings. The authors themselves noted that the accumulation of 15d-PGJ2 in senescent cells has been reported in various cell types, including human fibroblasts, HEPG2 hepatocellular carcinoma cells, and HUVEC endothelial cells (PMCID: PMC8501892). Although the current study observed similar activation of 15d-PGJ2 in myoblasts, it appears to be additive rather than fundamentally novel. The covalent adduct of 15d-PGJ2 with Cys-184 of H-Ras was reported over 20 years ago (PMID: 12684535), and the biochemical principles of this interaction are likely universal across different cell types. The regulation of myogenesis by both HRas and 15d-PGJ2 has also been previously extensively reported (PMID: 2654809, 1714463, 17412879, 20109525, 11477074). The main conceptual novelty may lie in the connection between these points in myoblasts. But as discussed in another comment, the use of C2C12 cells as a model for senescence study is questionable due to the lack of the key regulator p16. The findings in C2C12 cells may not accurately represent physiological-relevant myoblasts. It is recommended that these findings be validated in primary myoblasts to strengthen the study's conclusions.

(2) The C2C12 cell line is not an ideal model for senescence study.

C2C12 cells are a well-established model for studying myogenesis. However, their suitability as a model for senescence studies is questionable. C2C12 cells are immortalized and do not undergo normal senescence like primary cells as C2C12 cells are known to have a deleted p16/p19 locus, a crucial regulator of senescence (PMID: 20682446). The use of C2C12 cells in published studies does not inherently validate them as a suitable senescence model. These studies may have limitations, and the appropriateness of the C2C12 model depends on the specific research goals.

In the study by Moustogiannis et al. (PMID: 33918414), they claimed to have aged C2C12 cells through multiple population doublings. However, the SA-β-gal staining in their data, which is often used to confirm senescence, showed almost fully confluent "aged" C2C12 cells. This confluent state could artificially increase SA-β-gal positivity, suggesting that these cells may not truly represent senescence. Moreover, the "aged" C2C12 cells exhibited normal proliferation, which contradicts the definition of senescence. Similar findings were reported in another study of C2C12 cells subjected to 58 population doublings (PMID: 21826704), where even at this late stage, the cells were still dividing every 2 or 3 days, similar to younger cells at early passages. More importantly, I do know how the p16 was detected in that paper since the locus was already mutated. In terms of p21, there was no difference in the proliferative C2C12 cells at day 0.

In the study by Moiseeva et al. in 2023 (PMID: 36544018), C2C12 cells were used for senescence modeling for siRNA transfection. However, the most significant findings were obtained using primary satellite cells or confirmed with complementary data.

In conclusion, while molecular changes observed in studies using C2C12 cells may be valid, the use of primary myoblasts is highly recommended for senescence studies due to the limitations and questionable senescence characteristics of the C2C12 cell line.

(3) Regarding source of increased PGD in the conditioned medium, I want to emphasize that it's unclear whether the PGD or its metabolites increase in response to DNA damage or the senescence state. Thus, using a different senescent model to exclude the possibility of DNA damage-induced increase will be crucial.

(4) Similarly for the in vivo Doxorubicin (Doxo) injection, both reviewers have raised concerns about the potential side effects of Doxo, including inflammation, DNA damage, and ROS generation. These effects could potentially confound the results of the study. The physiological significance of this study will heavily rely on the in vivo data. However, the in vivo senescence component is confounded by the side effects of Doxo.

(5) Figure 2A lacks an important control from non-senescent cells during the measurement of C2C12 differentiation in the presence of conditioned medium. The author took it for granted that the conditioned medium from senescent cells would inhibit myogenesis, relying on previous publications (PMID: 37468473). However, that study was conducted in the context of myotonic dystrophy type 1. To support the inhibitory effect in the current experimental settings, direct evidence is required. It would be necessary to include another control with conditioned medium from normal, proliferative C2C12 cells.

(6) Statistical analyses problems.

Only t-test was used throughout the study even when there are more than two groups. Please have a statistician to evaluate the replicates and statistical analyses used.

For the 15d-PGJ2/cell concentration measurements in Figure 1F, there were only two replicates, which was provided in the supplementary table after required. Was that experiment repeated with more biological replicates?

For figure 1C, Fig 1F, 1G, 1J, 2C, 2E, 3A, 3E, 3F, 4D, 4E, please include each data points in bar graphs as used in Fig 1D, or at least provide how many biological replicates were used for each experiment?

There is no error bar in a lot of control groups (Fig 2C, 2E, 3EF, 4E, S4B).

For qPCR data in Figure 1C, the author responded in that the data in was plotted using 2-ΔCT instead of 2-ΔΔCT to show the variability in the expression of mRNAs isolated from animals treated with Saline. This statement does not align with the method section. Please revise.

(7) For Figure 1, the title may not be appropriate as there is insufficient data to support the inhibition of myoblast differentiation.

---

## [Author Response]

The following is the authors’ response to the current reviews.

**Public Reviews:**

**Reviewer #1 (Public Review):**
Summary:The authors show that upon treatment with Doxorubicin (Doxo), there is an increase in senescence and inflammatory markers in the muscles. They also show these genes get upregulated in C2C12 myoblasts when treated with conditioned media or 15d-PGJ2. 15dPGJ2 induces cell death in the myoblasts, decreases proliferation (measured by cell numbers), and decreases differentiation and fusion. 15d-PGJ2 modified Cys184 of HRas, which is required for its activation as indicated by the FRET analysis with RAF RBD. They also showed that 15d-PGJ2 activates ERK signaling, but not Akt signaling, through the electrophilic center. 15d-PGJ2 inhibits Golgi localization of HRAS (only WT, not C181 or C184 mutant). They also showed that expressing the WT HRas followed by 15d-PGJ2 treatment led to a decrease in the levels of MHC mRNA and protein, and this defect is dependent on C184. This is a well-written manuscript with interesting insights into the mechanism of action of 15d-PGJ2. However, some clarification and experiments will help the paper advance the field significantly.Strengths:The data clearly shows that 15d-PGJ2 has a negative role in the myoblast cells and that it leads to modification of HRas protein. Moreover, the induction of biosynthetic enzymes in the PGD2 pathway also supports the induction of 15d-PGJ2 in Doxorubicin-treated cells. Both conditioned media experiments and the 15d-PGJ2 experiments show that 15d-PGJ2 could be the active component secreted by the senescent myoblasts.Weaknesses:The genes that are upregulated in the muscles upon injection with Doxo are also markers for inflammation. Since Doxo is also known to induce systemic inflammation, it is important to delineate these two effects (Inflammatory cells vs senescent cells). The expression of beta Gal and other markers of senescence in the tissue sections will help to delineate these.

As pointed out Doxo induces systemic inflammation along with inducing DNA damage-mediated senescence. Therefore, along with the inflammatory markers of the SASP (CXCL1/2, TNF1α, IL6, PTGS1/2, PTGDS) we also observed an increase in the mRNA levels of canonical markers of DNA damage-mediated senescence. We observed an increase in the mRNA levels of cell cycle and senescence associated proteins p16 and p21 (Fig. 1C). We also observed an increased nuclear accumulation of p21 (Fig. 1A) and increased levels of phosphorylated H2A.X in the nucleus (Fig. 1B).

In Figure 2, where the defect in the differentiation of myoblasts upon treatment with 15d-PGJ2 is shown, most of the cells die within 48 hours at higher concentrations, making it difficult to perform the experiments. This also shows that 15d-PGJ2 was toxic to these cells. Lower concentrations show a decrease in the differentiation based on the lower number of nuclei in fibers and low expression of MyoD, MyoG, and MHC. However, it is unclear if this is due to increased cell death or defective differentiation. It would be a lot more informative if the cell count, cell division, and cell death could be plotted for these concentrations of the drug during the experiment.

We measured the viability of C2C12 cells after 24 hours of treatment with 15d-PGJ2 using the MTT assay and observed that the viability of cells was decreased after treatment with 15d-PGJ2 (10 µM) but not with 15d-PGJ2 (1 µM, 2 µM, 4 µM, or 5 µM) (see Fig. S2A of the updated manuscript). The results and figures of the manuscript have been updated accordingly.

Also, in the myoblast experiments, are the effects of treatment with Dox reversible?

The treatment with Doxorubicin is irreversible as the senescent phenotype was not reversed after withdrawal of Doxorubicin, even after 20 days.

In Figure 3, most of the experiments are done at a high concentration, which induces almost complete cell death within 48 hours.

Figure 3 is an acute experiment for only 1 hour, at which time no cell death was observed. Specifically, we measured the phosphorylation of Erk and Akt proteins after 1 hour of treatment with 15d-PGJ2 (10 µM) during which we did not observe any cell death.

Even at such a high concentration of 15dPGJ2, the increase in ERK phosphorylation is minimal.

We observe a ~30% increase in the phosphorylation of Erk proteins after treatment with 15d-PGJ­2 in 0.2% serum medium compared to treatment with vehicle (DMSO). This is reproducible and significant.

The experiment Figure 4C shows that C181 and C84 mutants of the HRas show higher levels in Golgi compared with WT. However, this could very well be due to the defect in palmitoylation rather than the modification with 15d-PGJ2.

Our data does not suggest higher levels of C184S mutant in the Golgi compared with WT (Fig. S4A). We observed that the ratio of HRas levels in the Golgi to the HRas levels in the plasma membrane were similar in C2C12 cells expressing HRas C184S and HRas WT (Fig. S4A graph columns 1 and 5).

Though the authors allude to the possibility that intracellular redistribution of HRas by 15d-PGJ2 requires C181 palmitoylation, the direct influence of C184 modification on C181 palmitoylation is not shown. To have a meaningful conclusion, the authors need to compare the palmitoylation and modification with 15d-PGJ2.

Palmitoylation of HRas C181S is required for the localization of HRas at the plasma membrane. The inhibition of palmitoylation of C181, either by mutation (C181S) or treatment with protein palmitoyl transferase inhibitor (2-Bromopalmitate), results in the accumulation of HRas at Golgi(Rocks et al., 2005) (Fig. S4A). Modification of HRas at C184 by 15d-PGJ2 (Fig. 3A) could inhibit the palmitoylation of HRas at C181. However, our data does not support this hypothesis as modification of HRas WT by 15d-PGJ2 does not increase the level of HRas at the Golgi, like in the case of inhibition of cysteine palmitoylation due to C181S mutation.

To test if the inhibition of myoblast differentiation depends on HRas, they overexpressed the HRas and mutants in the C2C12 lines. However, this experiment does not take the endogenous HRAs into consideration, especially when interpreting the C184 mutant. An appropriate experiment to test this would be to knock down or knock out HRas (or make knock-in mutations of C184) and show that the effect of 15d-PGJ2 disappears.

Endogenous HRas (wild type) is present in the C2C12 cells overexpressing the EGFP-tagged HRas constructs. Therefore, we only observe a partial rescue in the differentiation after 15d-PGJ2 treatment in C2C12 cells expressing the C184S mutant (Fig. 4D and E). However, since HRas is expressed under high expression CMV promoter and in the absence of other regulatory elements, the overexpressed constructs do show a dominant effect over the endogenous HRas, showing cysteine mutant dependent inhibition of differentiation of myoblasts after treatment with 15d-PGJ2 (Fig. 4D and E).

Moreover, in this specific experiment, it is difficult to interpret without a control with no HRas construct and another without the 15d-PGJ2 treatment.

The mRNA levels of MyoD, MyoG, and MHC in C2C12 cells expressing HRas constructs after treatment with 15d-PGJ2 were normalized to the mRNA levels in C2C12 cells expressing corresponding constructs and were treated with vehicle (DMSO). mRNA levels in C2C12 cells treated with vehicle were not shown as they were normalized to 1. MHC protein levels in C2C12 cells expressing HRas constructs after 15d-PGJ2 treatment were normalized to that in C2C12 cells treated with vehicle (DMSO). Since the hypothesis to study the effect of HRas cysteine mutations on the differentiation of myoblasts after treatment with 15d-PGJ2, C2C12 cells expressing HRas WT serve as adequate control. Fig. 2 shows the effect of 15d-PGJ2 on muscle differentiation when HRas was not overexpressed.

Moreover, the overall study does not delineate the toxic effects of 15d-PGJ2 from its effect on the differentiation.

The inhibition of differentiation in C212 cells after treatment with 15d-PGJ2 cannot be attributed to the general toxicity of 15d-PGJ2 in cells. We show that the inhibition of differentiation of myoblasts after 15d-PGJ2 depends on modification of HRas at C184 i.e. failure to modify HRas at C184 (Fig. 3A) and resultant activation (Fig. 3B) by 15d-PGJ2 rescues this inhibition of differentiation of C2C12 cells (Fig. 4D and E), dissecting the inhibition of differentiation of myoblasts by 15d-PGJ2 from general toxic effects of 15d-PGJ2 on cell physiology.

Please note that the effect of 15d-PGJ2 on cell physiology is context-specific. On one hand, 15d-PGJ2 has been shown to exert tumor-suppressor effects by inhibiting the proliferation of ovarian cancer cells and lung adenocarcinoma cells (de Jong et al., 2011; Slanovc et al., 2024), 15d-PGJ2 also exerts pro-carcinogenic effects by induction of epithelial to mesenchymal transition in breast cancer cells MCF7 and inhibition of tumor-suppressor protein p53 in MCF7 and PC-3 cells (Choi et al., 2020; Kim et al., 2010).

**Reviewer #2 (Public Review):**
Summary:In this study, Swarang and colleagues identified the lipid metabolite 15d-PGJ2 as a potential component of senescent myoblasts. They proposed that 15d-PGJ2 inhibits myoblast proliferation and differentiation by binding and regulating HRas, suggesting its potential as a target for restoring muscle homeostasis post-chemotherapy.Strengths:The regulation of HRas by 15d-PGJ2 is well controlled.Weaknesses:(1) I still think the novelty is limited by previous published findings. The authors themselves noted that the accumulation of 15d-PGJ2 in senescent cells has been reported in various cell types, including human fibroblasts, HEPG2 hepatocellular carcinoma cells, and HUVEC endothelial cells (PMCID: PMC8501892). Although the current study observed similar activation of 15d-PGJ2 in myoblasts, it appears to be additive rather than fundamentally novel. The covalent adduct of 15d-PGJ2 with Cys-184 of H-Ras was reported over 20 years ago (PMID: 12684535), and the biochemical principles of this interaction are likely universal across different cell types. The regulation of myogenesis by both HRas and 15d-PGJ2 has also been previously extensively reported (PMID: 2654809, 1714463, 17412879, 20109525, 11477074). The main conceptual novelty may lie in the connection between these points in myoblasts. But as discussed in another comment, the use of C2C12 cells as a model for senescence study is questionable due to the lack of the key regulator p16. The findings in C2C12 cells may not accurately represent physiological-relevant myoblasts. It is recommended that these findings be validated in primary myoblasts to strengthen the study's conclusions.

This is the first study to show a molecular mechanism where activation of HRas signaling in skeletal myoblasts due to covalent modification by 15d-PGJ2 at C184 of HRas inhibits the differentiation of skeletal myoblasts.

(2) The C2C12 cell line is not an ideal model for senescence study.C2C12 cells are a well-established model for studying myogenesis. However, their suitability as a model for senescence studies is questionable. C2C12 cells are immortalized and do not undergo normal senescence like primary cells as C2C12 cells are known to have a deleted p16/p19 locus, a crucial regulator of senescence (PMID: 20682446). The use of C2C12 cells in published studies does not inherently validate them as a suitable senescence model. These studies may have limitations, and the appropriateness of the C2C12 model depends on the specific research goals.

Several reports have shown that cells undergo senescence independent of p16 expression. MCF7 human breast adenocarcinoma cells have been shown to undergo DNA damage mediated and Oncogene induced senescence as seen after treatment with Doxorubicin (PMID: PMC7025418) and expression of constitutively active HRas (PMID: 17135242), despite the homozygous deletion of p16 locus (ISBN 9780124375512 Chapter 17 Table 2) by upregulation of cell cycle inhibitor protein p21. In this study, we observe an increase in the senescence markers in C2C12 cells after treatment with Doxo (Fig. 1). We also observed an increase in the markers of DNA damage-mediated senescence in MCF7 after treatment with Doxo (Data will be included in the revised manuscript). Based on these observations, we have concluded that C2C12 cells undergo senescence despite lacking the p16/p19 locus.

In the study by Moustogiannis et al. (PMID: 33918414), they claimed to have aged C2C12 cells through multiple population doublings. However, the SA-β-gal staining in their data, which is often used to confirm senescence, showed almost fully confluent "aged" C2C12 cells. This confluent state could artificially increase SA-β-gal positivity, suggesting that these cells may not truly represent senescence. Moreover, the "aged" C2C12 cells exhibited normal proliferation, which contradicts the definition of senescence. Similar findings were reported in another study of C2C12 cells subjected to 58 population doublings (PMID: 21826704), where even at this late stage, the cells were still dividing every 2 or 3 days, similar to younger cells at early passages. More importantly, I do know how the p16 was detected in that paper since the locus was already mutated. In terms of p21, there was no difference in the proliferative C2C12 cells at day 0.In the study by Moiseeva et al. in 2023 (PMID: 36544018), C2C12 cells were used for senescence modeling for siRNA transfection. However, the most significant findings were obtained using primary satellite cells or confirmed with complementary data.In conclusion, while molecular changes observed in studies using C2C12 cells may be valid, the use of primary myoblasts is highly recommended for senescence studies due to the limitations and questionable senescence characteristics of the C2C12 cell line.(3) Regarding source of increased PGD in the conditioned medium, I want to emphasize that it's unclear whether the PGD or its metabolites increase in response to DNA damage or the senescence state. Thus, using a different senescent model to exclude the possibility of DNA damage-induced increase will be crucial.

Though Senescence can be induced by several stress stimuli like DNA damage, Oncogene expression, ROS, Mitochondrial Dysfunction, etc., DNA damage remains critical for the induction of the SASP (reviewed in PMID: 20078217). Also, other models of senescence, like Oncogene Induced Senescence (reviewed in PMID: 17671427), ROS Induced Senescence (PMID: 24934860), Mitochondrial Dysfunction Associated Senescence (MiDAS) (PMID: 26686024) have shown upregulation of DNA damage-associated signaling pathways. In this study, we have explored the SASP of cells undergoing senescence upon chemotherapy drug Doxorubicin-mediated DNA damage.

(4) Similarly for the in vivo Doxorubicin (Doxo) injection, both reviewers have raised concerns about the potential side effects of Doxo, including inflammation, DNA damage, and ROS generation. These effects could potentially confound the results of the study. The physiological significance of this study will heavily rely on the in vivo data. However, the in vivo senescence component is confounded by the side effects of Doxo.

We concur that this is a limitation of this study and the subsequent work will demonstrate the origin of prostaglandin biosynthesis after treatment with Doxo in vivo.

(5) Figure 2A lacks an important control from non-senescent cells during the measurement of C2C12 differentiation in the presence of conditioned medium. The author took it for granted that the conditioned medium from senescent cells would inhibit myogenesis, relying on previous publications (PMID: 37468473). However, that study was conducted in the context of myotonic dystrophy type 1. To support the inhibitory effect in the current experimental settings, direct evidence is required. It would be necessary to include another control with conditioned medium from normal, proliferative C2C12 cells.

Conditioned medium of senescent cells of several types, like senescent myoblasts in case of DM1 (PMID: 37468473), adipocytes undergoing senescence due to H2O2 treatment, Insulin Resistance, and Replicative senescence (PMID: 37321332), has been shown to inhibit the differentiation of myoblasts. Therefore, in this study, we measured the effect of prostaglandin PGD2 and its metabolites on the differentiation of myoblasts by inhibiting the biosynthesis of PGD2 in senescent myoblasts by treatment with AT-56. We inhibited the synthesis of PGD2 in senescent cells by treatment with AT-56, and then collected the conditioned medium. Conditioned medium collected from senescent C2C12 cells treated with vehicle (DMSO) served as a control for the experiment.

(6) Statistical analyses problems.Only t-test was used throughout the study even when there are more than two groups. Please have a statistician to evaluate the replicates and statistical analyses used.

In experiments with more than two groups, the t-test was used for column-wise comparison of the experiment samples to the control sample. Multiple sample comparisons using one-way or two-way ANOVA were avoided as experimental samples were individually compared to the control sample.

For the 15d-PGJ2/cell concentration measurements in Figure 1F, there were only two replicates, which was provided in the supplementary table after required. Was that experiment repeated with more biological replicates?

Additional replicates of the experiment will be included in the revised manuscript.

For figure 1C, Fig 1F, 1G, 1J, 2C, 2E, 3A, 3E, 3F, 4D, 4E, please include each data points in bar graphs as used in Fig 1D, or at least provide how many biological replicates were used for each experiment?

Appropriate revisions will be made in the figure legends of the revised manuscript.

There is no error bar in a lot of control groups (Fig 2C, 2E, 3EF, 4E, S4B).

There are no error bars for the control groups in the figures 2C, 2E, 3E, 3F, 4E, and S4B as the experimental samples of each replicate were normalized to the corresponding control sample, rendering the values for the control sample of each replicate to 1.

For qPCR data in Figure 1C, the author responded in that the data in was plotted using 2-ΔCT instead of 2-ΔΔCT to show the variability in the expression of mRNAs isolated from animals treated with Saline. This statement does not align with the method section. Please revise.

Appropriate revisions will be made to the method sections of the revised manuscript.

(7) For Figure 1, the title may not be appropriate as there is insufficient data to support the inhibition of myoblast differentiation.

Appropriate revisions will be made to the revised manuscript.

**Recommendations for the authors:**
After careful review, the editors advise you to carefully address the following concerns.(1) There were concerns that in the revised manuscript, the DMSO and Doxo experiments depicted in Figure 1H appeared quite homogenous despite the author's description to the contrary. This leads to concerns about the type of statistics employed and the possible low number of replicates of experiments shown in Fig. 1.(2) Experiments in Figure 1F, 1I, and 1J had as few as n=2 experiments. Figures 1C, 1D, 1F, 1G, and 1J, the statistics used a two-tailed student's t-test; for all other experiments, they marked N/A for statistics. Using a t-test for multi-group comparisons (as indicated in the figure legend) and relying on only 2 replicates for many experiments are not appropriate.

Additional replicates for the experiments shown in figures 1F, 1I, and 1J have been done and the data will be revised along with updated statistical tests during the revision of the manuscript.

(3) In several experiments, the difference between technical replicates is too high.
**Reviewer #1 (Recommendations For The Authors):**
Most of my concerns were addressed in the revised manuscript.

We thank the reviewer for their time in reviewing the manuscript and consideration of the author’s response to their comments in during the previous round of review.

**Reviewer #2 (Recommendations For The Authors):**
Validating the findings in a primary myoblast is highly recommended for senescence studies due to the limitations and questionable senescence characteristics of the C2C12 cell line.

We have explained the statistical tests used in the manuscript in the general comment section of the reviewer’s comments.

Validate the finding in a different senescent model to exclude the possibility of DNA damage-response.

We have explained the statistical tests used in the manuscript in the general comment section of the reviewer’s comments.

For Fig 2A, add another control with a conditioned medium from normal, proliferative C2C12 cells.

We have explained the statistical tests used in the manuscript in the general comment section of the reviewer’s comments.

Please have a statistician to evaluate the replicates and statistical analyses used.

We have explained the statistical tests used in the manuscript in the general comment section of the reviewer’s comments.

For the barplots (figure 1C, Fig 1F, 1G, 1J, 2C, 2E, 3A, 3E, 3F, 4D, 4E), please include each data points, or at least provide how many biological replicates were used for each experiment.

Appropriate revisions will be made in the figure legends of the revised manuscript.

For Figure 1, the title may not be appropriate as there is insufficient data to support the inhibition of myoblast differentiation.

Appropriate revisions will be made to the revised manuscript.

The following is the authors’ response to the original reviews.

**eLife assessment**
This manuscript provides useful information about the lipid metabolite 15d-PGJ2 as a potential regulator of myoblast senescence. The authors provide experimental evidence that 15d-PGJ2 inhibits myoblast proliferation and differentiation by binding and regulating HRas. However, the manuscript is incomplete in its current form, as it lacks robust support from the data regarding the main conclusions related to senescence and technical concerns related to the senescence models used in this study.

We are grateful to the editors and the reviewers for their time and comments in sharpening the science and the writing of the manuscript. We have attached a detailed response to emphasize that the manuscript does include robust evidence regarding the claims, which could have been missed during the review process. We have provided a better context for these points now.

**Public Reviews:**

**Reviewer #1 (Public Review):**
Summary:The authors show that upon treatment with Doxorubicin (Doxo), there is an increase in senescence and inflammatory markers in the muscles. They also show these genes get upregulated in C2C12 myoblasts when treated with conditioned media or 15d-PGJ2. 15dPGJ2 induces cell death in the myoblasts, decreases proliferation (measured by cell numbers), and decreases differentiation and fusion. 15d-PGJ2 modified Cys184 of HRas, which is required for its activation as indicated by the FRET analysis with RAF RBD. They also showed that 15d-PGJ2 activates ERK signaling, but not Akt signaling, through the electrophilic center. 15d-PGJ2 inhibits Golgi localization of HRAS (only WT, not C181 or C184 mutant). They also showed that expressing the WT HRas followed by 15d-PGJ2 treatment led to a decrease in the levels of MHC mRNA and protein, and this defect is dependent on C184. This is a well-written manuscript with interesting insights into the mechanism of action of 15d-PGJ2. However, some clarification and experiments will help the paper advance the field significantly.Strengths:The data clearly shows that 15d-PGJ2 has a negative role in the myoblast cells and that it leads to modification of HRas protein. Moreover, the induction of biosynthetic enzymes in the PGD2 pathway also supports the induction of 15d-PGJ2 in Doxorubicin-treated cells. Both conditioned media experiments and the 15d-PGJ2 experiments show that 15d-PGJ2 could be the active component secreted by the senescent myoblasts.Weaknesses:The genes that are upregulated in the muscles upon injection with Doxo are also markers for inflammation. Since Doxo is also known to induce systemic inflammation, it is important to delineate these two effects (inflammatory cells vs senescent cells). The expression of beta Gal and other markers of senescence in the tissue sections will help to delineate these.

As pointed out Doxo induces systemic inflammation along with inducing DNA damage-mediated senescence. Therefore, along with the inflammatory markers of the SASP (CXCL1/2, TNF1α, IL6, PTGS1/2, PTGDS) we also observed an increase in the mRNA levels of canonical markers of DNA damage-mediated senescence. We observed an increase in the mRNA levels of cell cycle and senescence associated proteins p16 and p21 (Fig. 1C). We also observed an increased nuclear accumulation of p21 (Fig. 1A) and increased levels of phosphorylated H2A.X in the nucleus (Fig. 1B).

In Figure 2, where the defect in the differentiation of myoblasts upon treatment with 15d-PGJ2 is shown, most of the cells die within 48 hours at higher concentrations, making it difficult to perform the experiments. This also shows that 15d-PGJ2 was toxic to these cells. Lower concentrations show a decrease in the differentiation based on the lower number of nuclei in fibers and low expression of MyoD, MyoG, and MHC. However, it is unclear if this is due to increased cell death or defective differentiation. It would be a lot more informative if the cell count, cell division, and cell death could be plotted for these concentrations of the drug during the experiment.

We measured the viability of C2C12 cells after 24 hours of treatment with 15d-PGJ2 using the MTT assay and observed that the viability of cells was decreased after treatment with 15d-PGJ2 (10 µM) but not with 15d-PGJ2 (1 µM, 2 µM, 4 µM, or 5 µM) (see Fig. S2A of the updated manuscript). The results and figures of the manuscript have been updated accordingly.

Also, in the myoblast experiments, are the effects of treatment with Dox reversible?

The treatment with Doxorubicin is irreversible as the senescent phenotype was not reversed after withdrawal of Doxorubicin, even after 20 days.

In Figure 3, most of the experiments are done at a high concentration, which induces almost complete cell death within 48 hours.

Figure 3 is an acute experiment for only 1 hour, at which time no cell death was observed. Specifically, we measured the phosphorylation of Erk and Akt proteins after 1 hour of treatment with 15d-PGJ2 (10 µM) during which we did not observe any cell death.

Even at such a high concentration of 15dPGJ2, the increase in ERK phosphorylation is minimal.

We observe a ~30% increase in the phosphorylation of Erk proteins after treatment with 15d-PGJ2 in 0.2% serum medium compared to treatment with vehicle (DMSO). This is reproducible and significant.

The experiment Figure 4C shows that C181 and C84 mutants of the HRas show higher levels in Golgi compared with WT. However, this could very well be due to the defect in palmitoylation rather than the modification with 15d-PGJ2.

Our data does not suggest higher levels of C184S mutant in the Golgi compared with WT (Fig. S4A). We observed that the ratio of HRas levels in the Golgi to the HRas levels in the plasma membrane were similar in C2C12 cells expressing HRas C184S and HRas WT (Fig. S4A graph columns 1 and 5).

Though the authors allude to the possibility that intracellular redistribution of HRas by 15d-PGJ2 requires C181 palmitoylation, the direct influence of C184 modification on C181 palmitoylation is not shown. To have a meaningful conclusion, the authors need to compare the palmitoylation and modification with 15d-PGJ2.

Palmitoylation of HRas C181S is required for the localization of HRas at the plasma membrane. The inhibition of palmitoylation of C181, either by mutation (C181S) or treatment with protein palmitoyl transferase inhibitor (2-Bromopalmitate), results in the accumulation of HRas at Golgi(Rocks et al., 2005) (Fig. S4A). Modification of HRas at C184 by 15d-PGJ2 (Fig. 3A) could inhibit the palmitoylation of HRas at C181. However, our data does not support this hypothesis as modification of HRas WT by 15d-PGJ2 does not increase the level of HRas at the Golgi, like in the case of inhibition of cysteine palmitoylation due to C181S mutation.

To test if the inhibition of myoblast differentiation depends on HRas, they overexpressed the HRas and mutants in the C2C12 lines. However, this experiment does not take the endogenous HRAs into consideration, especially when interpreting the C184 mutant. An appropriate experiment to test this would be to knock down or knock out HRas (or make knock-in mutations of C184) and show that the effect of 15d-PGJ2 disappears.

Endogenous HRas (wild type) is present in the C2C12 cells overexpressing the EGFP-tagged HRas constructs. Therefore, we only observe a partial rescue in the differentiation after 15d-PGJ2 treatment in C2C12 cells expressing the C184S mutant (Fig. 4D and E). However, since HRas is expressed under high expression CMV promoter and in the absence of other regulatory elements, the overexpressed constructs do show a dominant effect over the endogenous HRas, showing cysteine mutant dependent inhibition of differentiation of myoblasts after treatment with 15dPGJ2 (Fig. 4D and E).

Moreover, in this specific experiment, it is difficult to interpret without a control with no HRas construct and another without the 15d-PGJ2 treatment.

The mRNA levels of MyoD, MyoG, and MHC in C2C12 cells expressing HRas constructs after treatment with 15d-PGJ2 were normalized to the mRNA levels in C2C12 cells expressing corresponding constructs and were treated with vehicle (DMSO). mRNA levels in C2C12 cells treated with vehicle were not shown as they were normalized to 1. MHC protein levels in C2C12 cells expressing HRas constructs after 15d-PGJ2 treatment were normalized to that in C2C12 cells treated with vehicle (DMSO). Since the hypothesis to study the effect of HRas cysteine mutations on the differentiation of myoblasts after treatment with 15d-PGJ2, C2C12 cells expressing HRas WT serve as adequate control. Fig. 2 shows the effect of 15dPGJ2 on muscle differentiation when HRas was not overexpressed.

Moreover, the overall study does not delineate the toxic effects of 15d-PGJ2 from its effect on the differentiation.

The inhibition of differentiation in C212 cells after treatment with 15d-PGJ2 cannot be attributed to the general toxicity of 15d-PGJ2 in cells. We show that the inhibition of differentiation of myoblasts after 15d-PGJ2 depends on modification of HRas at C184 i.e. failure to modify HRas at C184 (Fig. 3A) and resultant activation (Fig. 3B) by 15d-PGJ2 rescues this inhibition of differentiation of C2C12 cells (Fig. 4D and E), dissecting the inhibition of differentiation of myoblasts by 15d-PGJ2 from general toxic effects of 15d-PGJ2 on cell physiology.

Please note that the effect of 15d-PGJ2 on cell physiology is context-specific. On one hand, 15d-PGJ2 has been shown to exert tumor-suppressor effects by inhibiting the proliferation of ovarian cancer cells and lung adenocarcinoma cells (de Jong et al., 2011; Slanovc et al., 2024), 15d-PGJ2 also exerts pro-carcinogenic effects by induction of epithelial to mesenchymal transition in breast cancer cells MCF7 and inhibition of tumor-suppressor protein p53 in MCF7 and PC-3 cells (Choi et al., 2020; Kim et al., 2010).

**Reviewer #2 (Public Review):**
Summary:In this study, Swarang and colleagues identified the lipid metabolite 15d-PGJ2 as a potential component of senescent myoblasts. They proposed that 15d-PGJ2 inhibits myoblast proliferation and differentiation by binding and regulating HRas, suggesting its potential as a target for restoring muscle homeostasis post-chemotherapy.Strengths:The regulation of HRas by 15d-PGJ2 is well controlled.Weaknesses:The novelty of the study is compromised as the activation of PGD and 15d-PGJ2, as well as the regulation of HRas and cell proliferation, have been previously reported.

Literature does not support this statement, and it is important to clarify this misimpression for the field as a whole.

Let us clarify-

Covalent modification of HRas by 15d-PGJ2 has been reported only twice in the literature(Luis Oliva et al., 2003; Yamamoto et al., 2011) in fibroblasts and neurons respectively.

Interaction between Hras and 15d-PGJ2 in skeletal muscles has not been shown before, even though both Hras and 15d-PGJ2 are shown to be key regulators of muscle homeostasis.

Activation of Hras by 15d-PGJ2 was reported first by Luis Oliva et al (Luis Oliva et al., 2003). However, this study does not comment on the functional implications of activation of Hras signaling.

Recently, our lab contributed to a study where the functional implication of activation of Hras signaling due to covalent modification by 15d-PGJ2 was shown in the maintenance of senescence phenotype (Wiley et al., 2021).

15d-PGJ2 was shown to inhibit the differentiation of myoblasts by Hunter *et al* (Hunter et al., 2001). This study hypothesized that the inhibition of myoblast differentiation is via 15d-PGJ2 mediated activation of the PPARγ signaling, the study also showed inhibition of myoblast differentiation independent of PPARγ activity, suggesting the presence of other mechanisms.

This is the first study to show a molecular mechanism where activation of Hras signaling in skeletal myoblasts due to covalent modification by 15d-PGJ2 at C184 of Hras inhibits the differentiation of skeletal myoblasts.

Additionally, there are major technical concerns related to the senescence models, limiting data interpretation regarding the relevance to senescent cells.Major concerns:(1) The C2C12 cell line is not an ideal model for senescence study due to its immortalized nature and lack of normal p16 expression. A more suitable myoblasts model is recommended, with a more comprehensive characterization of senescence features.

C2C12 is a good model for DNA damage-based senescence that is used in this manuscript. Several reports in the literature have shown the induction of senescence in C2C12 cells. Moiseeva et al 2023 show induction of senescence in C2C12 cells after etoposide-mediated DNA damage. Moustogiannis et al 2021 show the induction of replicative senescence in C2C12 cells. In this study, we show that C2C12 cells undergo DNA damage-mediated senescence after treatment with Doxo. We measured the induction of senescence in C2C12 cells upon DNA damage using several physiological (Nuclear Size, Cell Size, and SA β-gal) and molecular markers (mRNA levels of p21 and SASP factors (IL6 and TGFβ), protein levels of p21) of senescence (see Fig. 1 of the updated manuscript). The results and the figures in the manuscript have been updated accordingly.

(2) The source of increased PGD or its metabolites in the conditioned medium is unclear. Including other senescence models, such as replicative or oncogeneinduced senescence, would strengthen the study.

Fig. 1E shows time-dependent increase in the expression of PGD2 biosynthetic enzymes in senescent C2C12 cells. Fig. 1F shows an increase in the levels of 15dPGJ2 secreted by senescent C2C12 cells in the conditioned medium. This data shows that senescent C2C12 cells are the source of PGD and its metabolites in the conditioned medium.

Again, C2C12 is not suitable for replicative senescence due to its immortalized status.

We and others have shown that C2C12 cells undergo senescence, and this manuscript only used DNA damage induced senescence.

(3) In the in vivo part, it is unclear whether the increased expression of PTGS1, PTGS2, and PTGDS is due to senescence or other side effects of DOXO.

We concur that this is a limitation of this study and the subsequent work will demonstrate the origin of prostaglandin biosynthesis after treatment with Doxo in vivo.

(4) Figure 2A lacks an important control from non-senescent cells during the measurement of C2C12 differentiation in the presence of a conditioned medium.

Figure 2A tests the effect of prostaglandin PGD2 and its metabolites secreted by the senescent cells on the differentiation of myoblasts. Therefore, we inhibited the synthesis of PGD2 in senescent cells by treatment with AT-56, and then collected the conditioned medium. Conditioned medium collected from senescent C2C12 cells treated with vehicle (DMSO) served as a control for the experiment, whereas differentiation of C2C12 cells without any treatment serves as a positive control.

There is no explanation of how differentiation was quantified or how the fusion index was calculated.

The fusion index was calculated using a published myotube analyzer software (Noë et al., 2022). Appropriate information has been added to the materials and methods section of the manuscript.

**Recommendations for the authors:**

**Reviewer #1 (Recommendations For The Authors):**
Line 3: Expand SA in "SA β-gal".

The manuscript has been updated accordingly (See line 3).

Line 68: HRas is highly regulated by lipid modifications.

The manuscript has been updated accordingly (See line 67).

FiguresFigure S1A seemed incomplete (maybe some processing issue).

The Figure has been updated in the revised manuscript (See Fig. S1A).

Figure S1B-H are mislabeled.

The figure has been updated in the revised manuscript (See Fig. S1C, D, E, and F).

Figures S1E-H are not mentioned in the manuscript.

The manuscript has been updated accordingly (See line 120).

Many supplementary figures are not cited in the article.

The manuscript has been updated accordingly. (See lines 85, 120, 123, 166, 225, 356, 364, 412, and 413)

**Reviewer #2 (Recommendations For The Authors):**
(1) Clarify the injection method for Doxorubicin in B6J mice on line 83 (IP or IM).

Mice were injected intraperitoneally with Doxorubicin (as mentioned in the materials and methods, see lines 83 and 794)

(2) Address missing information in figures or figure legends.There is missing piece in Sup Fig 1A.

The figure has been updated in the revised manuscript (See Fig. S1A).

Correct labels in Sup Fig 1C and 1D.

The figure has been updated in the revised manuscript (See Fig. S1C, D, E, and F).

How would the authors explain the dramatic differences in the morphology of C2C12 cells treated with DOXO between bright field and SA-beta-gal staining images in Sup Fig 1B and 1C.

The SA β-gal image after treatment with Doxo does show a flattened cell morphology. Another field of view from the same experiment has been added in the figure to show the difference in the cell morphology more prominently in the revised manuscript (See Fig. 1H).

Provide explanations for Sup Fig 1E-1G, including the meaning of the y-axis and the blue dots and red lines.

We have provided an explanation for the multiple reaction monitoring mass spectrometry used to measure the concentration of 15d-PGJ2 in the conditioned medium in the revised manuscript (see lines 119-130 and the legends of Fig. S1C, D, and E)

(3) Please review the calculation of qPCR data in Figure 1C for correctness, ensuring reference samples with an average expression level of 1.

The data in Fig. 1C was plotted using 2-ΔCT instead of 2-ΔΔCT to show the variability in the expression of mRNAs isolated from animals treated with Saline.

(4) Please explain the calculation of 15d-PGJ2/cell concentration in Figure 1F and provide raw data for review, considering the substantial changes and small error bars. The method or result section lacks an explanation of how this calculation was performed. Additionally, there is no mention of the cell number count.

All the raw values (concentration of 15d-PGJ2 measured using mass spec and cell numbers counted at the time of collection of conditioned medium) are provided in the supplementary table 1. The standard curve to calculate the concentration of 15dPGJ2 in the conditioned medium is shown in Fig. S1F. The cell number was counted after trypsinization using a hemocytometer on the day of collection of the conditioned medium.

(5) Please clarify how cell number normalization and doubling time calculation were done in Fig 2B. Consider replacing the figure with a growth curve showing confluence on the y-axis for easier interpretation.

Cells were counted every 24 hours and the normalization was done to the number of cells counted on day 0 of the treatment (to consider attaching efficiency and other cell culture parameters). Doubling time was calculated as the reciprocal of the slope of the graph of log2(normalized cell number) vs time.